# Unlocking the potential of allogeneic Vδ2 T cells for ovarian cancer therapy through CD16 biomarker selection and CAR/IL-15 engineering

Derek Lee [1,9], Zachary Spencer Dunn [1,2,9], Wenbin Guo [3,9], Carl J. Rosenthal[1], Natalie E. Penn[1], Yanqi Yu [1], Kuangyi Zhou[1], Zhe Li[1], Feiyang Ma [4], Miao Li[1], Tsun-Ching Song[1], Xinjian Cen [1], Yan-Ruide Li[1], Jin J. Zhou[5], Matteo Pellegrini [3,4,6], Pin Wang [2] & Lili Yang [1,6,7,8] ✉

Allogeneic Vγ9Vδ2 (Vδ2) T cells have emerged as attractive candidates for developing cancer therapy due to their established safety in allogeneic contexts and inherent tumor-fighting capabilities. Nonetheless, the limited clinical success of Vδ2 T cell-based treatments may be attributed to donor variability, short-lived persistence, and tumor immune evasion. To address these constraints, we engineer Vδ2 T cells with enhanced attributes. By employing CD16 as a donor selection biomarker, we harness Vδ2 T cells characterized by heightened cytotoxicity and potent antibody-dependent cell-mediated cytotoxicity (ADCC) functionality. RNA sequencing analysis supports the augmented effector potential of Vδ2 T cells derived from CD16 high (CD16$^{Hi}$) donors. Substantial enhancements are further achieved through CAR and IL-15 engineering methodologies. Preclinical investigations in two ovarian cancer models substantiate the effectiveness and safety of engineered CD16$^{Hi}$ Vδ2 T cells. These cells target tumors through multiple mechanisms, exhibit sustained in vivo persistence, and do not elicit graft-versus-host disease. These findings underscore the promise of engineered CD16$^{Hi}$ Vδ2 T cells as a viable therapeutic option for cancer treatment.

The recent success of chimeric antigen receptor (CAR)-T cell therapies in treating hematological malignancies highlights the transformative potential of genetically engineered cell therapies[1–3]. CARs are fusion proteins linking a targeting moiety, typically the single chain variable

fragment of an antibody, to T cell stimulatory domains, allowing CAR-engineered cells to both target and eliminate cancer cells[4]. Notably, conventional αβ T cells serve as the foundation for six FDA approved CAR-engineered cell products, with applications targeting

[1]Department of Microbiology, Immunology & Molecular Genetics, University of California, Los Angeles, CA, USA. [2]Mork Family Department of Chemical Engineering and Materials Science, University of Southern California, Los Angeles, CA, USA. [3]Bioinformatics Interdepartmental Program, University of California, Los Angeles, CA, USA. [4]Department of Molecular, Cell, and Developmental Biology, University of California, Los Angeles, CA, USA. [5]Department of Biostatistics, Fielding School of Public Health, University of California, Los Angeles, CA, USA. [6]Eli and Edythe Broad Center of Regenerative Medicine and Stem Cell Research, University of California, Los Angeles, CA, USA. [7]Jonsson Comprehensive Cancer Center, David Geffen School of Medicine, University of California, Los Angeles, CA, USA. [8]Molecular Biology Institute, University of California, Los Angeles, CA, USA. [9]These authors contributed equally: Derek Lee, Zachary Spencer Dunn, Wenbin Guo. ✉e-mail: liliyang@ucla.edu

CD19 for the treatment of B cell malignancies and B-cell maturation antigen (BCMA) for multiple myeloma. Although CAR-T therapies have been actively explored in solid tumor contexts, their therapeutic benefits have been limited[5], despite encouraging results from GD2- and Claudin18.2-targeting CAR-T cells in small patient cohorts[6,7]. Solid tumors present several challenges to CAR-T cells, including infiltration barriers, antigen heterogeneity, and immunosuppressive tumor microenvironments (TME)[8]. In addition to solid tumor efficacy concerns, CAR-T cells can cause severe adverse events, such as cytokine release syndrome (CRS), and their autologous nature restricts their accessibility[9]. The complexity of their manufacturing process and the reliance on patient-derived starting materials also result in exorbitant costs, time constraints, and product variability.

Allogeneic cell therapies offer potential solutions to address limitations of the present-day CAR-T cell paradigm[10]. Extensive research with αβ T cells, along with their clinical validation in autologous products, has spurred active investigations into developing allogeneic αβ CAR-T cells. To circumvent the risk of graft-versus-host disease (GvHD) arising from the recognition of mismatched MHC molecules by endogenous αβ TCRs, gene editing is frequently employed. CAR-engineered antigen-specific αβ T cells, such as CMV-specific T cells, is another approach to avoid GvHD and create off-the-shelf αβ CAR-T cells. Other strategies revolve around innate and innate-like immune cell populations that inherently pose lower risks for allogeneic transfer, such as macrophages, natural killer (NK) cells, invariant natural killer T (iNKT) cells, and γδ T cells[11,12].

γδ T cells, expressing T cell receptors (TCRs) composed of γ and δ chains, represent a unique subset, constituting only 1–10% of circulating T cells but exhibit conserved antimicrobial and antitumor functions[13]. γδ T cells display characteristics bridging the innate and adaptive immune systems, mediating cytotoxicity through both TCR and NK receptor signaling and expressing an array of context-dependent immunomodulatory cytokines[14,15]. Their cancer-fighting potential has been underscored by studies indicating that tumor-infiltrating γδ T cells have the most favorable prognostic impact among all immune cell subsets in various hematological and solid cancers[16]. In addition, as non-peptide-MHC dependent responders, γδ T cells are not expected to cause GvHD, making them safe for clinical use in allogeneic settings[17].

Upwards of 90% of peripheral γδ T cells possess a Vγ9Vδ2 TCR, which senses elevated phosphorylated nonpeptide metabolites, or phosphoantigens (pAg)[18]. Specifically, dysregulation of the mevalonate metabolism causes an accumulation of intracellular pAgs, which results in conformational changes in B7-related membrane protein butyrophilin (BTN) 3A1. This alteration allows BTN3A1 to interact with BTN2A1, forming a complex that is recognized by the Vγ9Vδ2 TCR[19,20]. Dysregulated cellular energetics, an emerging hallmark of cancer[21], empowers TCR-dependent killing of a wide range of liquid and solid tumors by Vγ9Vδ2 T cells[22]. Additionally, the activation of these cells can be achieved through bisphosphonates, a class of drugs that inhibit farnesyl pyrophosphate synthase in the mevalonate metabolic pathway, leading to an accumulation of pAgs in treated cells and the subsequent activation of the Vγ9Vδ2 TCR. Notable examples of such drugs include Zoledronate (ZOL) and Pamidronate (PAM).

Despite the favorable attributes that encourage the development of Vγ9Vδ2 (Vδ2) T cells as cell carriers for off-the-shelf CAR therapies, the literature on CAR-Vδ2 T cells is relatively limited, especially when compared to the wealth of research on conventional αβ T cells. Furthermore, engineering therapeutic cells with transgenic IL-15, which has been shown to enhance the preclinical efficacy of CAR-engineered NK[23], iNK[24], NKT[25], and Vδ1 T[26] cells, has not been reported for CAR-Vδ2 T cells.

CD16 (FcγRIII) is a well-established IgG receptor known for mediating antibody-dependent cell-mediated cytotoxicity (ADCC). Previous studies have investigated the presence and functionality of CD16 on Vδ2 T cells[27–30]. In this work, we introduce a straightforward method for generating Vδ2 T cells with enhanced antitumor activity. This process involves screening donors for CD16 expression on Vδ2 T cells. Vδ2 T cells obtained from CD16[Hi] donors exhibit phenotypic traits advantageous for cancer treatment, including increased expression of effector molecules and ADCC activity. Additionally, we employ mesothelin-targeted CAR and IL-15 engineering to further enhance the antitumor potential of CD16[Hi] Vδ2 T cells. Our results highlight the feasibility, therapeutic potential, and high safety profile of engineered CD16[Hi] Vδ2 T cells in the context of cancer treatment.

## Results

### CD16 serves as a biomarker to screen PBMC donors for high performance Vδ2 T cells

The expansion of Vγ9Vδ2 (Vδ2) T cells from a large cohort of peripheral blood mononuclear cell (PBMC) donors for the development of γδ T cell-based cancer therapies revealed notable differences in CD16 expression (Fig. 1a–d; Supplementary Fig. 1a, b). The initial CD16 expression on PBMC-derived Vδ2 T cells (before stimulation) ranged from nearly lack of CD16 expression to over 35%. CD16 high (CD16[Hi]) Vδ2 T cells and CD16 low (CD16[Lo]) Vδ2 T cells were defined as ≥ 35% CD16 expression and ≤20% CD16 expression, respectively. A total of 30 healthy donors were screened for this experiment, in which 7 (23.3%) were classified as CD16[Hi] donors (Fig. 1b). Notably, CD16 expression on CD16[Hi] Vδ2 T cells was not only maintained but also increased upon Vδ2 T cell activation with ZOL and expansion for 14 days (Fig. 1c, d). Using the ZOL and IL-2 expansion method, we produced comparable expansion of Vδ2 T cells irrespective of CD16 expression (Fig. 1e).

Past studies have shown that Vδ2 T cells exert potent cytotoxicity against various types of tumors. We compared the killing of human ovarian tumor cells by Vδ2 T cells expanded from CD16[Hi] and CD16[Lo] donors. High-grade serous ovarian cancer cell lines, OVCAR3 and SKOV3, were engineered with firefly luciferase and green fluorescence protein dual reporters (FG) and cocultured with various ratios of effector cells (effector to tumor, E:T ratio) in the presence or absence of ZOL (Fig. 1f, g). Twenty-four hours post co-culture, tumor cell killing was measured by bioluminescence; three CD16[Hi] and three CD16[Lo] Vδ2 T cell donors were used. In the presence of ZOL, CD16[Hi] Vδ2 T cells displayed significantly enhanced cytotoxicity at almost all the E:T ratios tested, for both cancer cell lines. The improved cytotoxicity of Vδ2 T cells expanded from CD16[Hi] donors correlated with increased IFN-γ secretion, as measured by ELISA, and perforin and granzyme B production, as measured by intracellular staining, following 24 h (h) co-culture of cancer cells and effector cells at a 1:1 E:T ratio in the presence or absence of ZOL (Fig. 1h). Depending on the cancer cell type and assay used, Vδ2 T cells are capable of tumor killing as well as cytokine, perforin, and granzyme B production in the absence of ZOL; in some cases, ZOL or other preconditioning is used to exert effective cancer killing[31,32]. However, for the ovarian cancer cells we tested, Vδ2 T cells exhibited a dependency on ZOL for cytotoxicity and effector molecule production during in vitro cocultures (Fig. 1h). Although differences in cytotoxicity potential were observed, the expression of chemokine receptors CXCR3, CCR4, and CCR5 was comparable within donors and between donors, whereas CD56 was upregulated on CD16[+] cells within donors and expressed at higher overall levels on Vδ2 T cells from CD16[Hi] donors. CCR2 was upregulated on CD16[-] cells within donors and expressed at overall higher levels on Vδ2 T cells from CD16[Lo] donors (Supplementary Fig. 1c, d). Importantly, the expression of granzyme B and perforin was similar between CD16[+] and CD16[-] Vδ2 T cells within a donor, and both types of cells displayed higher expression levels in Vδ2 T cells expanded from CD16[Hi] donors than those expanded from

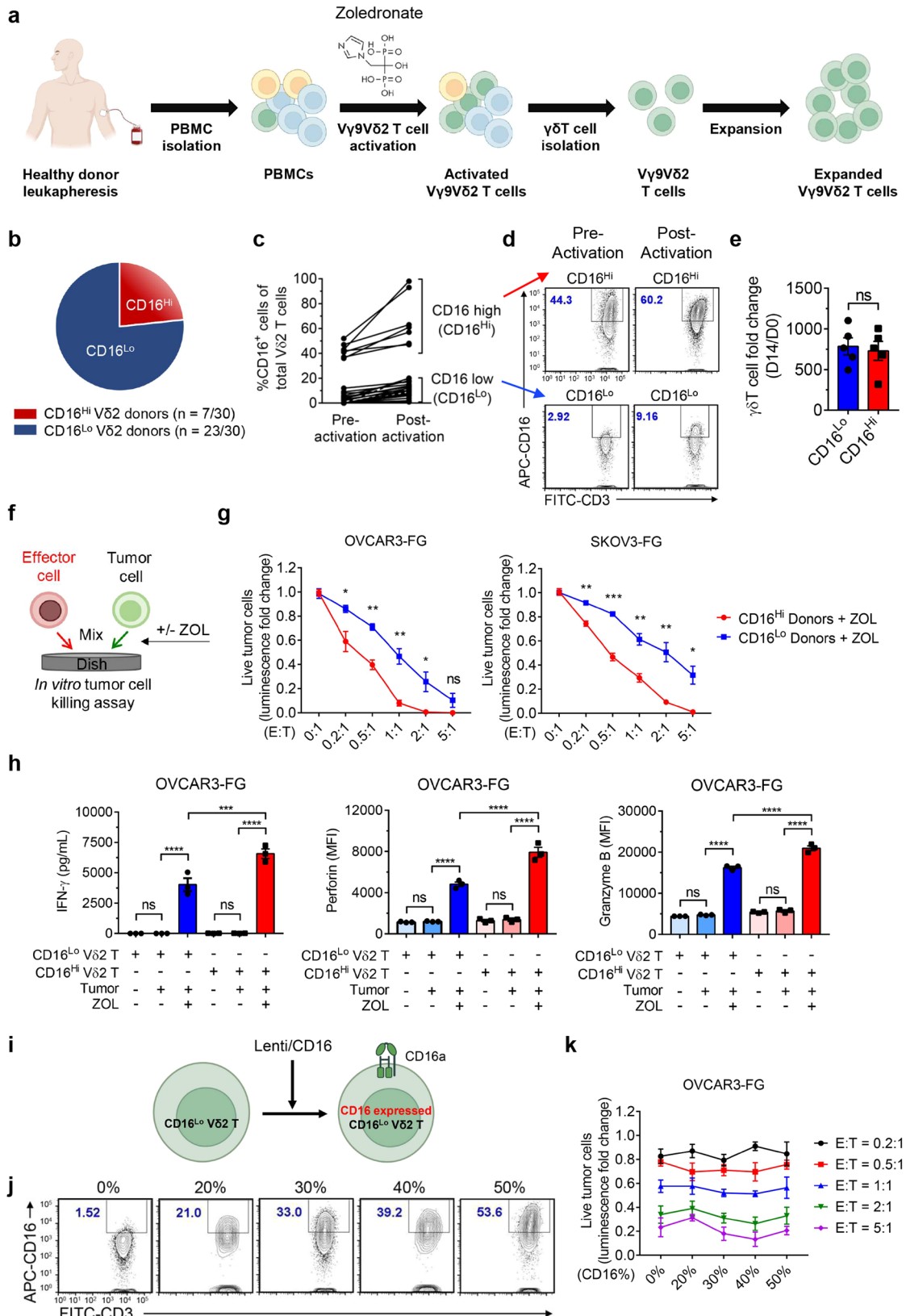

CD16$^{Lo}$ donors (Supplementary Fig. 1e–h). These results indicate that Vδ2 T cells from CD16$^{Hi}$ donors may be a favorable cell type for developing Vδ2 T cell-based cancer therapies.

We then created a lentiviral vector encoding CD16a[33] (Lenti/CD16) for engineering CD16$^{Lo}$ Vδ2 T cells to express transgenic CD16a (Fig. 1i, j). Importantly, titration of CD16 expression did not affect cytotoxicity during in vitro cocultures with OVCAR3-FG cells (Fig. 1k). This suggests that CD16 could be used as a biomarker to select for donors with highly potent Vδ2 T cells rather than functioning as an active receptor that enhances tumor killing, and that genetic introduction of CD16 to CD16$^{Lo}$ Vδ2 T cells may not recapitulate the heightened activity of Vδ2 T cells expanded from the CD16$^{Hi}$ donors.

**Fig. 1 | CD16 serves as a biomarker to screen peripheral blood mononuclear cell (PBMC) donors for high performance Vδ2 T cells. a** Experimental design to generate PBMC-derived Vγ9Vδ2 T (referred to as Vδ2 T) cells. Zoledronate (ZOL) and IL-2 were used to activate and expand Vδ2 T cells. **b** Pie chart showing the proportions of CD16 high (CD16$^{Hi}$) and CD16 low (CD16$^{Lo}$) Vδ2 T cell donors. Note, a total of 30 healthy donors were screened. The cutoff between CD16$^{Hi}$ and CD16$^{Lo}$ donors was 35% of CD16$^+$ cells out of total Vδ2 T cells. **c** FACS quantification of % CD16$^+$ cells of total Vδ2 T cells before and after activation and expansion. PBMCs from 30 different donors were used. Note, Vδ2 T cells were expanded for 10 - 14 days. **d** Representative FACS plots of (**c**). **e** Expansion of CD16$^{Hi}$ and CD16$^{Lo}$ Vδ2 T cells over 14 days (*n* = 5; *n* indicates different donors). **f–h** Studying the in vitro antitumor efficacy of CD16$^{Hi}$ and CD16$^{Lo}$ Vδ2 T cells in the presence or absence of ZOL. Two human ovarian cancer cell lines were studied as the tumor targets: OVCAR3-FG and SKOV3-FG; both cancer cell lines were engineered to express the firefly luciferase and green fluorescence protein (FG) dual reporters. **f** Experimental design. **g** Tumor cell killing data at 24 h (*n* = 3 from 3 different donors). **h** ELISA measurements of IFN-γ production (left), and FACS measurements of intracellular perforin (middle) and granzyme B (right) production, at 24 h (E:T ratio = 1:1; *n* = 3 from 3 different donors). **i–k** Studying the overexpression of CD16 in CD16$^{Lo}$ Vδ2 T cells. CD16$^{Lo}$ Vδ2 T cells were transduced with a lentivector encoding the human CD16 gene (Lenti/CD16). **i** Schematic design of overexpression experiment. **j** FACS detection of CD16 overexpression on CD16$^{Lo}$ Vδ2 T cells transduced with titrated amounts of Lenti-CD16. **k** In vitro killing of OVCAR3-FG cells by Lenti/CD16 transduced CD16$^{Lo}$ Vδ2 T cells in the presence of ZOL at 24 h after co-culture (*n* = 3). Representative of >10 (**a–d**), 5 (**e**), 3 (**f–h**), and 1 (**i–k**) experiments. Data are presented as the mean ± SEM. ns not significant; *\*p* < 0.05; *\*\*p* < 0.01; *\*\*\*p* < 0.001; *\*\*\*\*p* < 0.0001 by Student's *t* test (**e**, **g**) or by one-way ANOVA (**h**). Source data and exact p values are provided as a Source Data file.

## CD16$^{Hi}$ Vδ2 T cells display enhanced cytotoxic gene signatures

To further assess the differences between Vδ2 T cells expanded from different donors, we performed bulk RNA-Seq on Vδ2 T cells expanded from 13 PBMC donors, 3 of which were classified as CD16$^{Hi}$ based on our flow cytometry criteria. CD16 mRNA expression was assessed (Fig. 2a), and the samples were grouped into CD16$^{Hi}$ and CD16$^{Lo}$ Vδ2 T cells using k-means clustering algorithm (k = 2). Three Vδ2 T cell samples were identified as CD16$^{Hi}$ and the rest were CD16$^{Lo}$, confirming our flow cytometry results. Principal-component analysis (PCA) showed the clustering of CD16$^{Hi}$ T cells, potentially indicating that there is transcriptomic variance between Vδ2 T cells correlating with CD16 expression (Fig. 2b). Differential expression analysis based on CD16 expression revealed an upregulation of genes associated with effector functions, such as *GNLY*, *CD86*, and *CX3CR1*, in high CD16 expressors (Fig. 2c). Various genes related to antitumor effector functions, such as genes encoding transcription factors, activation/homing markers, and cytotoxic molecules, were also assessed (Supplementary Fig. 2a). CD16 expression positively correlated with the expression of granzymes (*GZMA*, *GZMB*), perforin (*PRF1*), and natural killer receptors (*NCR1*). Interestingly, the CD16$^{Hi}$ group exhibited a downregulation of the transcription factor-encoding gene *RORC*, which is associated with a Th17-like phenotype. Th17-like Vδ2 T cells have been shown to promote cancer progression in several syngeneic cancer models and there is evidence for their detrimental effects in human malignancies[34–36].

Furthermore, gene set enrichment analysis (GSEA)[37], were performed to characterize the biological pathway signatures associated with CD16 expression (Fig. 2d and Supplementary Fig. 2b). High CD16 expression enriched for signatures associated with immune effector functions and activation, such as cytotoxicity, degranulation, Fc gamma receptor signaling, and phagocytosis. There was also an enrichment for proliferation-related genes in the CD16$^{Hi}$ samples (Fig. 2d), although a difference in expansion was not observed during Vδ2 T cell in vitro production (Fig. 1e). A gene ontology (GO) over-representation Cnet plot[38] displaying the linkages of differentially expressed genes and biological processes confirmed the increase in immune cell activation signatures in CD16$^{Hi}$ Vδ2 T cells (Fig. 2e).

We also assessed the relationship between CD16 expression in Vδ2 T cell samples and the gene signatures of 24 immune cell types defined in ImmuCellAI[39] (Fig. 2f). Enrichment scores of the immune cell types for each sample were calculated using single-sample GSEA (ssGSEA) implemented in the Gene Set Variation Analysis (GSVA) package[40]. The resulting correlations between enrichment scores and the samples' CD16 expression level show that CD16 expression enriched for cytotoxic, NK, and macrophage signatures as well as an absence of Tfh and Th17 signatures. CD16 expression also corresponded to a Th1 signature, which is supported by the heightened secretion of IFN-γ by CD16$^{Hi}$ Vδ2 T cells. In summary, the bulk RNA-Seq results align with the potent in vitro activity of CD16$^{Hi}$ Vδ2 T cells and provide future directions for cell product characterization, especially with regard to IL-17 production and expansion potential.

## Development of CAR- and IL-15-engineered CD16$^{Hi}$ Vδ2 T cells targeting mesothelin for the treatment of ovarian cancer

Mesothelin (MSLN) is a cell-surface glycoprotein with limited expression on normal tissue but is overexpressed in many solid cancers, including ovarian, lung, and pancreatic carcinomas, making MSLN a promising target for cancer therapies, including CAR-engineered cell therapies[41,42]. Due to the heterogenous expression of MSLN in ovarian cancer, we hypothesized that MSLN-targeting CAR (MCAR)-engineered Vδ2 T cells may exhibit superior antitumor activity due to the potential of multiple targeting capability. We also implemented cell engineering to produce IL-15 (termed MCAR15), as IL-15 signaling has been shown to enhance the persistence of innate/innate-like immune cells, and, to the best of our knowledge, has not yet been explored to modulate CAR Vδ2 T cells.

Following the schematic in Fig. 3a, we produced MCAR and MCAR15-engineered CD16$^{Hi}$ Vδ2 T cells (MCAR-Vδ2T and MCAR15-Vδ2T) and included non-engineered CD16$^{Hi}$ Vδ2 T cells as the control (NT-Vδ2T). MCAR and MCAR15 constructs resulted in similar CAR expression, Vδ2 T cell expansion, and Vδ2 T cell purity, with greater than 98% purity routinely achieved (Fig. 3b–g). MCAR15-Vδ2T cells expressed higher levels of persistence-associated proteins, pSTAT5, BcL-xL, and BcL-2 as measured by intracellular staining and flow cytometry (Fig. 3h). Marked IL-15 secretion was seen in the activated MCAR15 group (Fig. 3i) upon coculture with OVCAR3-FG cells, which is consistent with previously published data that demonstrate increased IL-15 production by CAR/IL-15 engineered αβ T[43], Vδ1 T[26], and NK cells[23] following antigen stimulation. This may be due to the heightened metabolic activity and protein translation that occur during cell activation as well as the short half-life of IL-15. Although various definitions of Vδ2 T cell memory status do exist in literature[44,45], our analysis based on CD27 and CD45RA expression showed that both CAR15$^+$ and CAR15$^-$ populations from the MCAR15-Vδ2T cell group were mostly central memory (~40%) and effector memory (~ 50%) phenotypes (Fig. 3j–l).

## MCAR15-Vδ2T cells demonstrate robust in vitro antitumor activity against multiple ovarian cancer cell lines

We conducted in vitro cytotoxicity and cytokine production assays to evaluate the effector functions of CAR-engineered CD16$^{Hi}$ Vδ2 T cells (Fig. 4). A third ovarian cancer cell line expressing FG dual reporter, OVCAR8-FG, was also included for several assays. The MSLN expression on OVCAR3-FG, OVCAR8-FG, and SKOV3-FG was assessed (Fig. 4b), revealing variable MSLN expression in the three ovarian cancer models. The conventional αβ CAR T cells targeting MSLN (MCAR-T) cells were included as a control, and Vδ2 T cell groups were added with and without ZOL unless otherwise specified. Following 24 h cocultures, all the effector cell groups exhibited efficient killing of OVCAR3-FG cancer cells (Fig. 4c). However, against OVCAR8-FG cells,

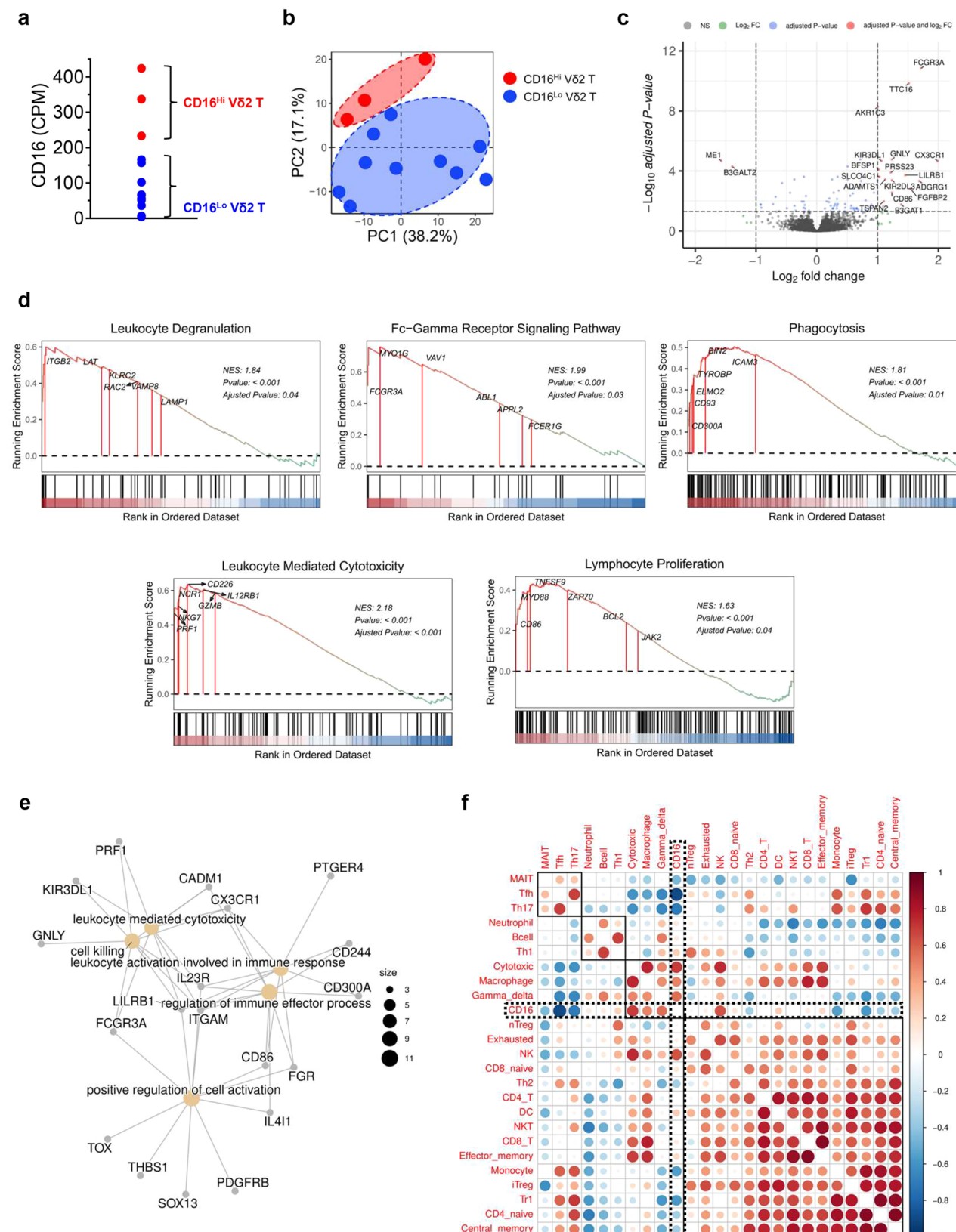

Vδ2 T cell groups displayed heightened cytotoxicity compared to MCAR-T cells, and only in the presence of ZOL were Vδ2 T cells able to kill CAR-antigen negative SKOV3-FG cells (Fig. 4c). No differences were observed between the two engineered Vδ2 T cells groups (with and without IL-15). In parallel, we assessed the intracellular expression of granzyme B and perforin by MCAR15-Vδ2T cells co-cultured with OVCAR3-FG and SKOV3-FG cells in the presence or absence of ZOL (Fig. 4d). We observed an upregulation of effector molecule production in the presence of OVCAR3-FG cells with and without ZOL, whereas ZOL was required to increase effector molecule production in cocultures with SKOV3-FG cells. We also compared the in vitro cytotoxicity of CAR and IL-15 engineered CD16^Hi and CD16^Lo Vδ2 T cells and

**Fig. 2 | CD16^Hi Vδ2 T cells display enhanced cytotoxic gene signatures.** Vδ2 T cells expanded from 13 PBMC donors were subject to RNA-Seq analysis. **a** Dot plot showing the CD16 mRNA expression in Vδ2 T cells (CPM: counts per million reads mapped). The samples were categorized into CD16^Hi ($n = 3$) and CD16^Lo ($n = 10$) Vδ2 T cells using k-means clustering algorithm (k = 2). **b** Principal-component analysis (PCA) plot showing the coordination of CD16^Hi and CD16^Lo Vδ2 T cells. **c** Volcano plot showing the differential gene expression between CD16^Hi and CD16^Lo Vδ2

T cells. FC, fold change. NS not significant. **d** Gene set enrichment analysis (GSEA) plots showing a significant enrichment of indicated gene signatures in CD16^Hi Vδ2 T cells. NES, normalized enrichment score. **e** Cnet plot showing the enriched GO pathways. **f** Correlation plot showing the relationship between CD16 expression in Vδ2 T cell samples and immune cell type scores. Circle size and color intensity are proportional to correlation; red and blue represent positive and negative correlations, respectively. Representative of 1 experiment.

witnessed significantly improved killing by the CD16^Hi group (Supplementary Fig. 3a, b).

Twenty-four hour cocultures were also used to monitor the IFN-γ production (Fig. 4e). When cultured with OVCAR3-FG, CAR-engineered T and Vδ2 T cells secreted ample IFN-γ, whereas non-engineered control Vδ2 T cells did not, highlighting the benefit of CAR-engineering. In corroboration with the cytotoxicity findings, IFN-γ was only produced in response to SKOV3-FG by CAR-Vδ2 T cells in the presence of ZOL. The cytotoxicity and IFN-γ results illustrate the tumor cell multi-targeting potential by CAR-Vδ2 T cells through both CAR and TCR recognition.

We performed in vitro repeated tumor challenge assays to investigate the long-term functionality of CAR15- and CAR-Vδ2 T cells (Fig. 4f, g). On days 0, 3, 6, 9, 12, and 15, effector cells were challenged with new tumor cells with and without ZOL in 96-well plates. Twenty-four hours following cancer cell addition, one of the 96-well plates was used for bioluminescence measurements to determine cancer killing. IL-15 secretion clearly improved cancer killing ability of MCAR15 Vδ2 T cell groups in repeated tumor challenges against OVCAR3-FG and OVCAR8-FG cells.

To further illustrate CAR-antigen dependent and independent antitumor activity, we generated MSLN-negative OVCAR3-FG (^KOOVCAR3-FG) cells using CRISPR-Cas9 editing (Fig. 4h). ^KOOVCAR3-FG cells were cultured with CAR-engineered effector cells in repeated tumor challenges assays following the schematic shown in Fig. 4f. MSLN knockout resulted in reduced killing by CAR-Vδ2 T cells in the absence of ZOL compared to with ZOL after the repeated tumor challenge (Fig. 4i), whereas no difference was seen in cytotoxicity towards the parental OVCAR3-FG cell line after 24 h cocultures (Fig. 4c). This indicates that in the absence of ZOL, the killing of parental OVCAR3-FG is driven by CAR-mediated killing. By day 7, after the 3^rd tumor challenge, ^KOOVCAR3-FG cell outgrowth occurred for all effector cell groups except for the MCAR- and MCAR15-Vδ2T cell cultures supplemented with ZOL. The MSLN knockout studies show that CAR-antigen presentation can enhance the killing functionality of CAR-Vδ2 T cells and CAR-antigen escape can potentially be overcome by the Vδ2 TCR-mediated killing mechanism.

## Engineered CD16^Hi Vδ2 T cells can target ovarian cancer cells through ADCC

For several anticancer monoclonal antibodies (mAb) such as cetuximab and trastuzumab, Fc-mediated immune effector function, ADCC, is a major mode of action to deplete tumor cells[46,47]. T and NK cell-mediated ADCC is predominantly attributed to the CD16a (FcγRIIIa) transmembrane receptor, which is expressed by many effector cells of the immune system, whereas CD16b (FcγRIIIb), a GPI-anchored protein, is exclusively expressed by neutrophils[48]. Although CD32 (FcγRI) and CD64 (FcγRII) are expressed at low levels on Vδ2 T cells and may contribute to ADCC, they are primarily implemented in myeloid-mediated ADCC[49,50]. We thus focus our Vδ2 T cell studies on the canonical lymphocyte CD16a receptor, which we refer to as CD16.

To test the ADCC capacity of unmodified and engineered CD16^Hi Vδ2 T cells, we performed in vitro tumor coculture assays with a preclinical anti-HER2 mAb analog to trastuzumab (Fig. 5a). The expression of HER2 on OVCAR3-FG, OVCAR8-FG, and SKOV3-FG was assessed by flow cytometry (Fig. 5b). For unmodified CD16^Hi Vδ2 T cells, against

OVCAR3-FG cells, isotype control had minimal impact on Vδ2 T cell cytotoxicity, whereas significant enhancement of tumor killing was seen with concentrations of anti-HER2 mAb as low as 0.1 μg/mL (Fig. 5c). The addition of anti-HER2 mAb also improved the killing efficacy of MCAR-Vδ2T cells, with and without IL-15 secretion, against OVCAR3-FG, OVCAR8-FG, and SKOV3-FG cells, whereas no benefit was observed for conventional MCAR-T cells (Fig. 5d–f). We confirmed that CD16^Lo Vδ2 T cells lack detectable ADCC functions (Supplementary Fig. 3c, d). Importantly, ADCC enabled efficient killing of MSLN-negative SKOV3-FG cells by CD16^Hi MCAR-Vδ2T cells (Fig. 5f), and this was accompanied by pronounced secretion of IFN-γ (Fig. 5g). Lastly, a repeated tumor challenge assay was performed against ^KOOVCAR3-FG cells in the presence or absence of anti-HER2 mAb (Fig. 5h). Antibody treatment greatly enhanced tumor killing by MCAR- and MCAR15-Vδ2T cells, although it was only the MCAR15-Vδ2T cells and antibody combination that established prolonged tumor control (Fig. 5i).

## MCAR15-Vδ2T cells can kill tumor-associated macrophages in vitro

Macrophages (Mφ) are large, innate immune cells that phagocytose target cells in response to infection or insult and recent work highlights the role macrophages play in both tumor elimination and progression[51–54]. While existing on a continuum, macrophage populations that prevent cancer growth and activate antitumor immunity are commonly referred to as M1-type, and those that promote cancer growth and potentiate immunosuppression as M2-type. Most cancers are primarily populated by M2-type macrophages, which represent a logical target for immunotherapies. We generated human monocyte-derived M2 macrophages by culturing PBMCs in the presence of macrophage colony-stimulating factor (M-CSF), resulting in monocyte-derived macrophages (MDM), followed by culture with IL-4 and IL-13 to produce M2-polarized Mφ (Supplementary Fig. 4a)[55]. FACS detection of CD11b and CD14 revealed successful production of macrophages and further characterization of M2 Mφ markers CD163 and CD206 confirmed M2 polarization (Supplementary Fig. 4b, c). MCAR15-Vδ2T cells were cultured with M2 Mφ in vitro and the resulting cytotoxicity was monitored by flow cytometry (Supplementary Fig. 4d). After a 24 h culture, at a 1:1 MCAR15-γδT: M2 Mφ, 50% of the Mφ cells were killed, and this was increased to 70% upon the addition of ZOL (Supplementary Fig. 4e). These results indicate that the killing of tumor-associated macrophages (TAMs) may be another mechanism that MCAR15-Vδ2T cells could potentially exploit to mediate antitumor immune reactivity.

## MCAR15-Vδ2T cells are safe and efficacious in intraperitoneal and subcutaneous in vivo ovarian cancer models

The in vivo antitumor activity and safety of MCAR15-Vδ2T cells derived from CD16^Hi donors were evaluated in two xenograft tumor models. In the first model, NSG mice were inoculated intraperitoneally (i.p.) with $1 \times 10^6$ OVCAR3-FG cells and fourteen days later the mice were treated with i.p. delivered $4 \times 10^6$ MCAR-T, MCAR-Vδ2T, or MCAR15-Vδ2T cells, or vehicle (PBS) (Fig. 6a). Tumor growth was monitored by live animal imaging using bioluminescence. Within one week of treatment, all the effector cell groups displayed significant tumor retardation compared to the control group and effective tumor control (Fig. 6b–d). Control mice reached endpoint around day 70 post-tumor injection due to

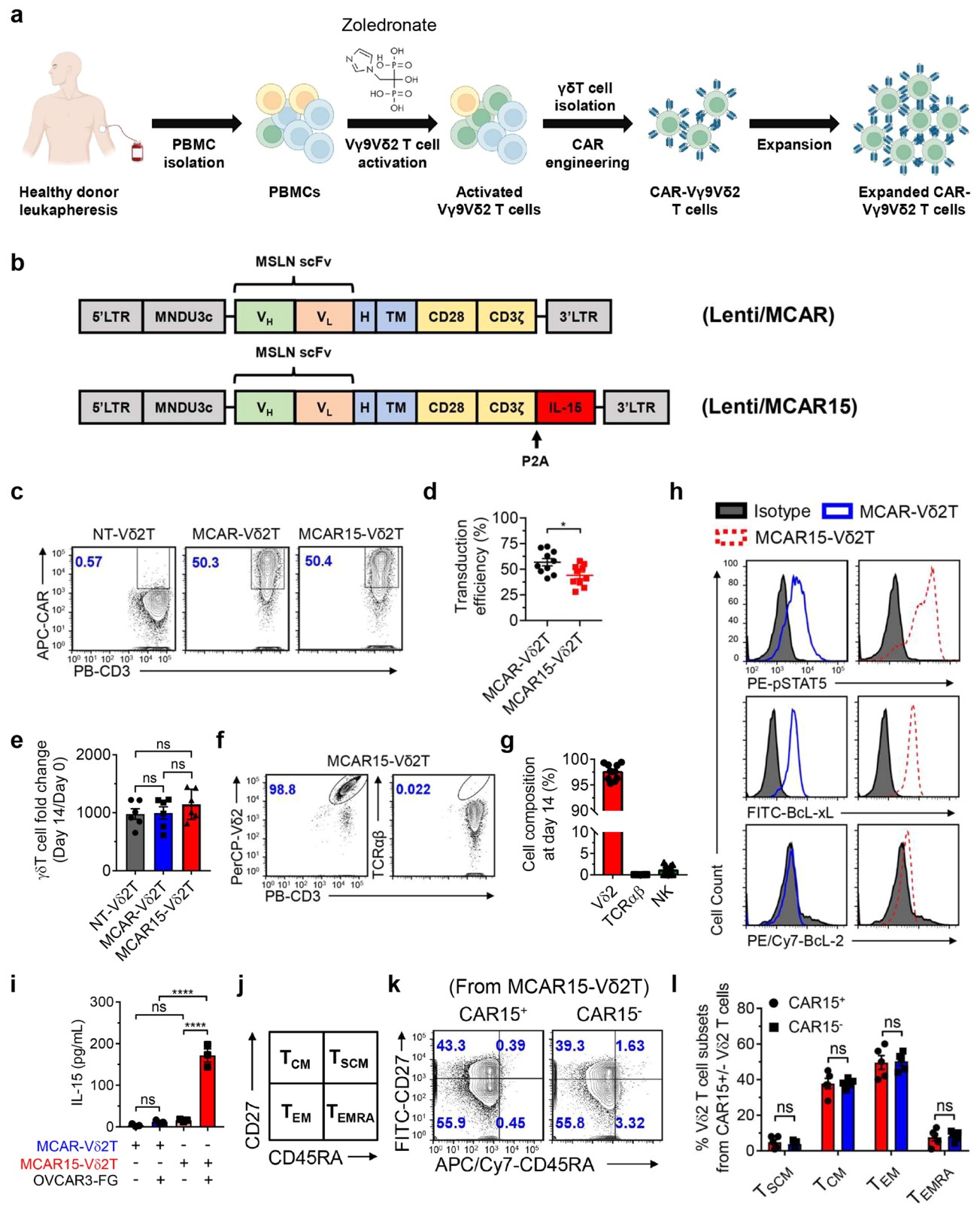

cancer burden, and MCAR-T mice perished shortly thereafter due to GvHD (as determined by weight loss, fur loss, and malaise) (Fig. 6e). For mice treated with MCAR-Vδ2T cells, 2/5 mice survived through the end of the study, day 180, with one complete elimination of the tumor and one mouse with tumor relapse, whereas the other three mice succumbed to relapsed tumors. MCAR15-Vδ2T cells resulted in complete remissions in all 5/5 mice through day 180 without any signs of

GvHD, highlighting the benefit of IL-15 engineering in long-term intraperitoneal tumor growth inhibition (Fig. 6e).

We then tested the therapeutic potential of MCAR15-Vδ2T cells in a subcutaneous tumor model. OVCAR8 subcutaneous tumors were established in NSG mice and, when the tumors reached an average of 50 mm³, $10 \times 10^6$ effector cells were administered intravenously (Fig. 7a). All the effector groups resulted in tumor growth inhibition,

**Fig. 3 | CD16^Hi Vδ2 T cells can be engineered with CAR/IL-15 while retaining expansion capacity and memory status. a** Experimental design to generate the MCAR/IL-15 engineered CD16^Hi Vδ2 T (MCAR15-Vδ2T) cells. Note, ZOL and IL-2 were used to activate and expand Vδ2 T cells. CAR, chimeric antigen receptor. **b** Schematic of the indicated lentivectors. Lenti/MCAR, lentivector encoding a mesothelin (MSLN)-targeting CAR (MCAR); Lenti/MCAR15: lentivector encoding the same MCAR as well as a secreting form of human IL-15. scFv: single-chain variable fragment; V_H: variable heavy chain; V_L: variable light chain; H: CD8 hinge; TM: CD28 transmembrane domain; CD28: CD28 intracellular domain; CD3ζ: CD3ζ intracellular domain; IL-15: interleukin 15. **c** FACS analysis of CAR expression on CD16^Hi Vδ2 T cells transduced with mock (denoted as non-transduced Vδ2 T cells; NT-Vδ2T cells), or with Lenti/MCAR (denoted as MCAR-Vδ2T cells), or with Lenti/MCAR15 (denoted as MCAR15-Vδ2T cells). **d** Quantification of c (n = 10; n indicates different donors). **e** Expansion of the indicated Vδ2 T cells over 14 days of culture (n = 5; n indicates different donors). **f** FACS analysis of the cellular composition of expanded MCAR15-Vδ2T cells. **g** Quantification of f (n = 10; n indicates different

donors), NK cells were gated as CD45^+CD56^+CD3^−. **h** FACS detection of IL-15 signaling events in MCAR15-Vδ2T cells. Phosphorylation of STAT5 (pSTAT5) and the upregulation of anti-apoptotic transcription factors Bcl-xL and BcL-2 were studied. Cells were cultured in cytokine free media for 48 h before intracellular staining. **i** ELISA measurements of IL-15 production from MCAR15-Vδ2T cells in the presence or absence of human ovarian cancer cell line OVCAR3-FG for 48 h (n = 3). **j**–**l** Studying the memory phenotype of MCAR15-Vδ2 T cells. **j** Schematic plot defining memory T cell subsets by the cell surface expression of CD27 and CD45RA biomarkers. Stem cell-like memory (T_SCM): CD27^+CD45RA^+; central memory (T_CM): CD27^+CD45RA^−; effector memory (T_EM): CD27^−CD45RA^−; terminally differentiated effector memory (T_EMRA): CD27^−CD45RA^+. **k** FACS detection of CD27 and CD45RA expression on CAR15^+ and CAR15^− MCAR15-Vδ2T cells. **l** Quantification of k (n = 5; n indicates different donors). Representative of 1 (**b**), 3 (**h**, **i**), 5 (**j**–**l**), and >10 (**c**–**g**) experiments. Data are presented as the mean ± SEM. ns, not significant; *p < 0.05; ****p < 0.0001 by Student's t test (**d**, **l**) or by one-way ANOVA (**e**, **i**). Source data and exact p values are provided as a Source Data file.

---

with MCAR15-Vδ2T cells sustaining significantly enhanced tumor control compared to the all the other groups by Day 57 (Fig. 7b, c). On Day 57, the mice were sacrificed for terminal analysis. Tumors were excised and processed to assess human immune cell infiltration. Mouse tissues were also harvested to assess effector cell persistence throughout the preclinical model as well as xenoreactivity (GvHD). MCAR15-Vδ2T cells displayed robust persistence, as shown by their increased presence in the tumor and all the mouse organs analyzed (Fig. 7d). The superior persistence was not associated with GvHD, as H&E-stained tissue sections collected from experimental mice at day 57 showed MCAR15-Vδ2T cells did not cause the accumulation of mononuclear cell infiltrates in the lung, liver, spleen, nor kidney, whereas MCAR-T cell-treated mice showed distinct manifestations of mononuclear cell aggregates (Fig. 7e). An additional OVCAR8 subcutaneous study showed that CD16^Hi MCAR15-Vδ2T cells exhibit superior in vivo tumor control compared to CD16^Lo MCAR15-Vδ2T cells (Supplementary Fig. 5). These in vivo results highlight the potential for CD16^Hi MCAR15-Vδ2T cells to treat intraperitoneal solid tumors without causing GvHD. The results also confirm the benefit of engineering CAR-Vδ2 T cells to secrete IL-15.

## Discussion

We present the development and characterization of CAR and IL-15 engineered CD16^Hi Vδ2 T cells as a promising avenue for the advancement of allogeneic cellular immunotherapies. We have identified CD16 as a valuable biomarker for selecting Vδ2 T cells with enhanced cytotoxicity. Engineered CD16^Hi Vδ2 T cells were generated at high yield and purity, targeted tumors via multiple mechanisms, such as CAR, TCR, and ADCC recognition. Furthermore, these cells demonstrated robust preclinical in vivo tumor control and long-lasting persistence without any signs of GvHD.

To overcome the current challenges of conventional CAR-based αβ T cell therapies, including limited efficacy against solid tumors, the engineering of innate-like and innate immune cells, such γδ T cells, iNKT cells, NK cells, and macrophages, is under active investigation[56–59]. Analogous to αβ T cells, innate/innate-like cell populations are heterogeneous mixtures of cells with different transcriptional programming, phenotype, and functionality, and certain subsets may be desirable for cell therapy against cancer. Laskowski et al. recently emphasized the inter-donor variability of NK cell profiles and the need for a thorough understanding of NK product characteristics to define biomarkers indicative of greater potency and persistence[60]. Our studies indicate that CD16 can potentially serve as a biomarker for the selection of Vδ2 T cell donors.

Vδ2 T cells are of high interest for developing cancer therapies given their notable safety in the allogeneic setting and intrinsic antitumor functions[22,61,62]. CAR engineering and altered culture conditions, such as TGF-β supplementation, have been preclinically studied to

enhance the therapeutic potential of Vδ2 T cells[63–65], although reported clinical investigation of modified Vδ2 T cells remains scarce[62]. Extensive research on CD16 (FcγRIIIa) and its role in tumor control through ADCC with both therapeutically administered[47,66,67] and naturally produced[68] antibodies prompted us to characterize CD16 expression on Vδ2 T cells isolated from healthy human peripheral blood mononuclear cells and assess the potential for CD16 to serve as a biomarker for donor selection. Further support for focusing on CD16 to create cancer immunotherapies comes from several avenues: the development of a high-affinity, non-cleavable CD16 that was incorporated into pluripotent stem cell-derived NK cells for improved antitumor capabilities when combined with mAbs[69], utilization of CD56^dimCD16^+ NK cells to improve dendritic cell vaccination response[70], and the bi- and tri-specific killer engagers (BiKEs and TriKEs) against tumor-specific antigens to enhance NK cell-mediated tumor rejection[71–74]. Our data suggests that CD16^Hi Vδ2 T cells can be another strategy for actualizing the potential of CD16-mediated antitumor immunity.

The expression and functionality of CD16 on Vδ2 T cells have been explored for over two decades[27–30]. One study speculated that the therapeutic potential of PBMC-derived CD16^+ Vδ2 T cells may be limited by their poor expansion capabilities[29] and a single cell RNA-Seq (scRNA-Seq) study indicated PBMC-derived CD16^+ Vδ2 T cells are fully differentiated[75], but other researchers demonstrated high yields of CD16^+ Vδ2 T cells[30]. In our hands, using traditional ZOL and IL-2 expansion methods, expanded CD16^Hi Vδ2 T cells had a similar proliferation and memory status compared to CD16^Lo Vδ2 T cells. Vδ2 T cells expanded from CD16^Hi donors had heightened cytotoxicity and cytokine production in response to ovarian cancer cell lines and deep RNA-Seq revealed activated, cytotoxic, and phagocytic gene signatures. We further developed CD16^Hi Vδ2 T cells for cancer therapy through CAR and IL-15 engineering and confirmed by in vitro and in vivo characterization that CAR15-Vδ2 T cells display robust antitumor efficacy against ovarian cancer models. Although IL-15 injections can increase circulating NK and CD8^+ T cells, achieving sustained IL-15 signaling using soluble IL-15 is difficult due to its short serum half-life and limited bioavailability[76]. IL-15 self-secretion has the potential to provide sustained and local delivery of IL-15 to engineered immune cells, as well as simplify treatment regimens and reduce systemic toxicity. In ovarian and other solid tumors, MSLN is a promising target, but antigen heterogeneity can curtail the effectiveness of single antigen-targeting modalities. Using CD16^Hi MCAR-Vδ2T cells, solid tumors can be killed through CAR- and TCR-mediated recognition as well as combination therapies with HER2 mAbs.

The Vγ9Vδ2 TCR recognizes dysregulated metabolism by binding conformational changes in BTN3A1 that result from altered mevalonate pathways, which commonly occurs in solid tumors and can license Vδ2 T cell killing[22,77]. Although the ovarian cancer cell lines used in our study required the addition of ZOL for Vδ2 TCR-mediated

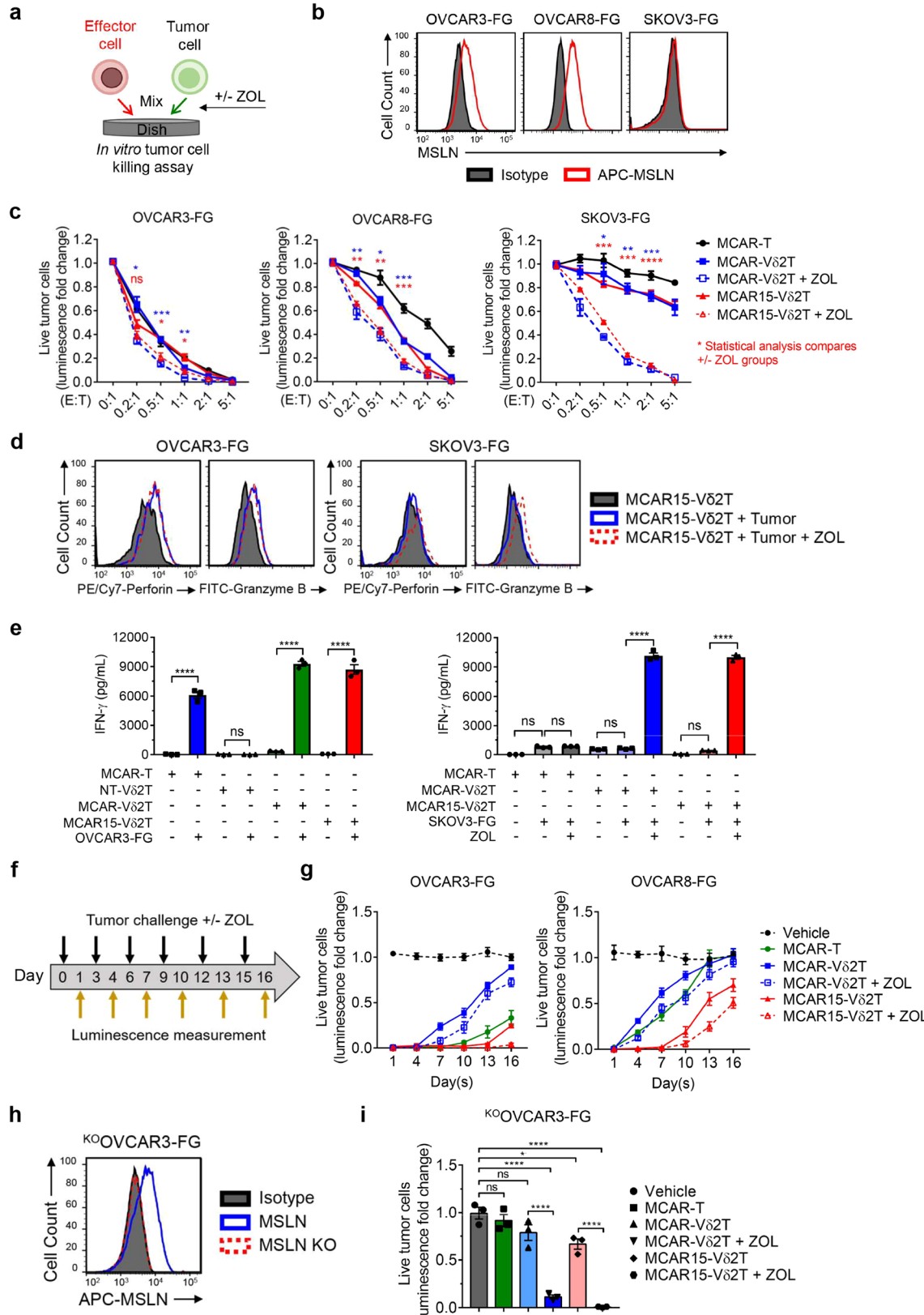

killing, deregulated cellular energetics is an emerging hallmark of cancer[21] and metabolic restrictions found in the tumor microenvironment may engender Vδ2 TCR targeting of tumor cells in the absence of exogenous bisphosphonates[31,32]. Another subset of γδ T cells under evaluation for cancer therapy bears the Vδ1 TCR, and allogeneic CD20-targeting CAR-Vδ1 T cell therapy (ADI-100) has a promising 75%

objective response rate (ORR) and 69% complete response (CR) rate in a small Phase I study ($n = 16$) of adults with relapsed/refractory advanced B-cell lymphoma[78]. Importantly, ADI-001 demonstrated a 100% CR rate ($n = 5$) in patients that relapsed after prior autologous anti-CD19 CAR T therapy. Vδ1 T cells mainly recognize glycolipids presented by MHC Class I-like CD1 proteins, which are predominantly

**Fig. 4 | MCAR15-Vδ2 T cells can effectively kill tumor cells via CAR/TCR dual-targeting mechanisms.** Three human ovarian cancer cell lines (OVCAR3-FG, OVCAR8-FG, and SKOV3-FG) and four effector cells (Vδ2T, MCAR-Vδ2T, MCAR15-Vδ2T, and MCAR-T) were used in the study. MCAR-T indicates conventional αβ T cells engineered to express the same MCAR (as a benchmark control). The same CD16$^{Hi}$ donor PBMCs were used to generate all 4 types of effector cells. **a**−**e** In vitro 24-hour tumor cell killing assay. **a** Experimental design. Tumor cells and effector cells were co-cultured for 24 h, then were subject to analysis. **b** FACS plots showing the expression of mesothelin (MSLN) tumor antigen on the indicated ovarian cancer cell lines. **c** Tumor cell killing data collected at 24 h ($n = 3$). **d** FACS detection of perforin and granzyme B intracellular production by MCAR15-Vδ2T cells at 24 h (E:T ratio = 1:1). **e** ELISA measurements of IFN-γ production at 24 h (E:T ratio = 1:1; $n = 3$). **f**−**i** In vitro repeated tumor cell challenge assay. **f** Experimental design. Effector cells were mixed with tumor cells and rechallenged every 3 days, and the tumor cell killing data were collected at 24 h after every dosing of tumor cells (E:T ratio = 2:1; $n = 3$). **g** Tumor cell killing data collected over time. **h** Generation and FACS validation of an MSLN-knockout OVCAR3-FG ($^{KO}$OVCAR3-FG) cell line. **i** $^{KO}$OVCAR3-FG tumor cell killing data collected after the 3rd tumor cell rechallenge. Representative of 3 (**a**, **c**−**g**, **i**), and 1 (**b**, **h**) experiments. Data are presented as the mean ± SEM. ns not significant; *$p < 0.05$; **$p < 0.01$; ***$p < 0.001$; ****$p < 0.0001$ by Student's $t$ test (**c**, **e**) or by one-way ANOVA (**i**). Source data and exact $p$ values are provided as a Source Data file.

expressed by antigen presenting cells[79–81]. Thus, to harness intrinsic TCR recognition of cancer cells, Vδ1 T cells are well-suited for hematological malignancies, whereas Vδ2 T cells can target both liquid and solid cancers. Other potential benefits of the Vδ2 subset include its stimulation by FDA-approved ZOL, higher starting cell number in peripheral blood (1–10% for Vδ2 vs 0.1–1% for Vδ1), and Vδ1 cells do not express CD16[82]. While NK-mediated enhancement of tumor-targeting Abs remains a focus[83], our results indicate CD16$^{Hi}$ Vδ2 T cells can be used to achieve antitumor ADCC. Both Vδ1 and Vδ2 subsets exploit NK-like activation and cytotoxicity through various natural killer receptors (NKRs), such as NKG2D and DNAM1, which can enhance the breadth and amplitude of their antitumor activity[82].

Our exploration into CAR and IL-15 engineered CD16$^{Hi}$ Vδ2 T cell therapy development revealed several areas for further interrogation to determine the potential of this population for cancer treatment, such as T cell memory status, in vivo polarization, immunogenicity, and CD16$^{Hi}$ Vδ2 T cell pool formation. Substantial efforts have focused on creating less differentiated, memory-like cells for CAR-based therapies[84,85]. Activation and expansion of T cells can result in differentiated final cell products, which can have reduced expansion potential. Interestingly, despite minimal starting numbers, expanded Vδ1 T cells maintain earlier memory status[26], which could be a benefit of Vδ1 T cells compared to Vδ2, which in our culture emerge predominantly as T effector memory (T$_{EM}$) cells. Methods for preserving the memory status of Vδ2 T cells during expansion without compromising expansion rates are under active investigation. Additional characterization, including ex vivo analysis following mouse tumor challenges, can be used to pressure-test the Th1 phenotype of CD16$^{Hi}$ Vδ2 T cell observed in RNA-Seq samples and confirm persistent functionality of the CD16 receptor. Although γδ T cells were identified as a prognostic marker for better outcomes[16] and recently shown to be effectors of immunotherapy in DNA mismatch repair-deficient cancers with HLA class I defects[86], other studies report the potential negative impact of IL-17-producing γδ T cells[34–36]. CD16$^{Hi}$ Vδ2 T cells express lower levels of ROR1 mRNA than their CD16$^{Lo}$ counterparts and single-sample GSEA indicated CD16 inversely correlates with Th17 polarization, but further assays will be needed to functionally examine Th17 potential in CD16$^{Hi}$ Vδ2 T cells.

It will also be important to address the immunogenicity of engineered CD16$^{Hi}$ Vδ2 T cells. Although lymphodepletion-based preconditioning is standard for adoptive cellular therapy[87,88], the question remains whether host-mediated rejection of the infused cells will affect their engraftment and obviate a therapeutic window to elicit meaningful clinical benefit. If immunogenicity were to critically hinder engineered CD16$^{Hi}$ Vδ2 T cells, haploidentical or HLA-matched donors or cloaking strategies, such as HLAI/II knockout and/or HLA-E overexpression, may be necessary to evade immune rejection.

Investigations into how the CD16$^{Hi}$ Vδ2 T cell pool is formed can deepen our understanding of CD16$^{Hi}$ Vδ2 T cells as a unique subpopulation. We have shown that transgenic CD16 on CD16$^{Lo}$ Vδ2 T cells did not affect cytotoxicity, supporting the notion that CD16 signifies more than a functional enhancer of Vδ2 T cell tumor killing. It was previously found that FcRγ-deficient NK (g-NK) cells exhibited

significantly more robust responsiveness upon stimulation through CD16, and subsequent studies revealed the g-NK cell pool is shaped and maintained in human cytomegalovirus (CMV)-infected individuals by a mechanism that involves both epigenetic modification and antibody-dependent expansion[89–91]. Additional analysis of CD16$^{Hi}$ Vδ2 T cells using scRNA-Seq and epigenomic sequencing may unveil their etiology and engender the creation of further enhanced CD16$^{Hi}$ Vδ2 T cell products.

There are several potential challenges to successful CD16$^{Hi}$ Vδ2 T cell-based therapy. Tumor penetration is an important consideration in developing CAR T cells to treat solid tumors, and while there is preliminary evidence that MCAR15-Vδ2T cells infiltrate subcutaneous tumors in vivo, additional studies and ultimately clinical investigation will be needed to show meaningful tumor penetration in patients. Upon infiltrating the tumor, sufficient persistence and antitumor functionality are significant clinical challenges, as immunorejection and the immunosuppressive tumor microenvironment may thwart CD16$^{Hi}$ Vδ2 T cell-based therapies. Furthermore, although CAR-engineered γδ T cell clinical trials have been conducted and are underway, the large-scale production of MCAR15-Vδ2T cells using CD16$^{Hi}$ donors will require significant manufacturing and process design optimization efforts. The findings from our study highlight the effectiveness of a combination of donor selection based on CD16 expression, CAR engineering, and IL-15 secretion in enhancing the cancer therapy potential of Vδ2 T cells. The high CD16 expression on Vδ2 T cells allows these cells to be used in combination with therapeutic antibodies, while their cytotoxicity towards M2-polarized macrophages provides additional antitumor properties. Given the heterogeneity of solid tumors and their complex immunosuppressive tumor microenvironment, a multi-faceted approach to tumor recognition and immunomodulation will likely be necessary to achieve long-lasting therapeutic results in treatment-resistant patients.

## Methods

### Mice

NOD.Cg-Prkdc$^{SCID}$Il2rg$^{tm1Wjl}$/SzJ (NOD/SCID/IL-2Rγ$^{-/-}$, NSG) mice were maintained in the animal facilities of the University of California, Los Angeles (UCLA) under the following housing conditions: temperature ranging from 68 °F to 79 °F, humidity maintained at 30% to 70%, a light cycle of On at 6:00 am and Off at 6:00 pm, and room pressure set to negative. 6–10 weeks old female mice were used for all experiments unless otherwise indicated. Due to ethical reasons, we terminated experiments when mice developed severe ascites or before tumor volume surpassed 1000 mm³. All animal experiments were approved by the Institutional Animal Care and Use Committee (IACUC) of UCLA. All mice were bred and maintained under specific pathogen-free conditions, and all experiments were conducted in accordance with the animal care and use regulations of the Division of Laboratory Animal Medicine (DLAM) at the UCLA.

### Medium, cytokines, and chemicals

Zoledronic acid monohydrate (ZOL) was purchased from Sigma (Cat. SML0223). Recombinant human IL-2, IL-4, IL-7, IL-15, IL-17,

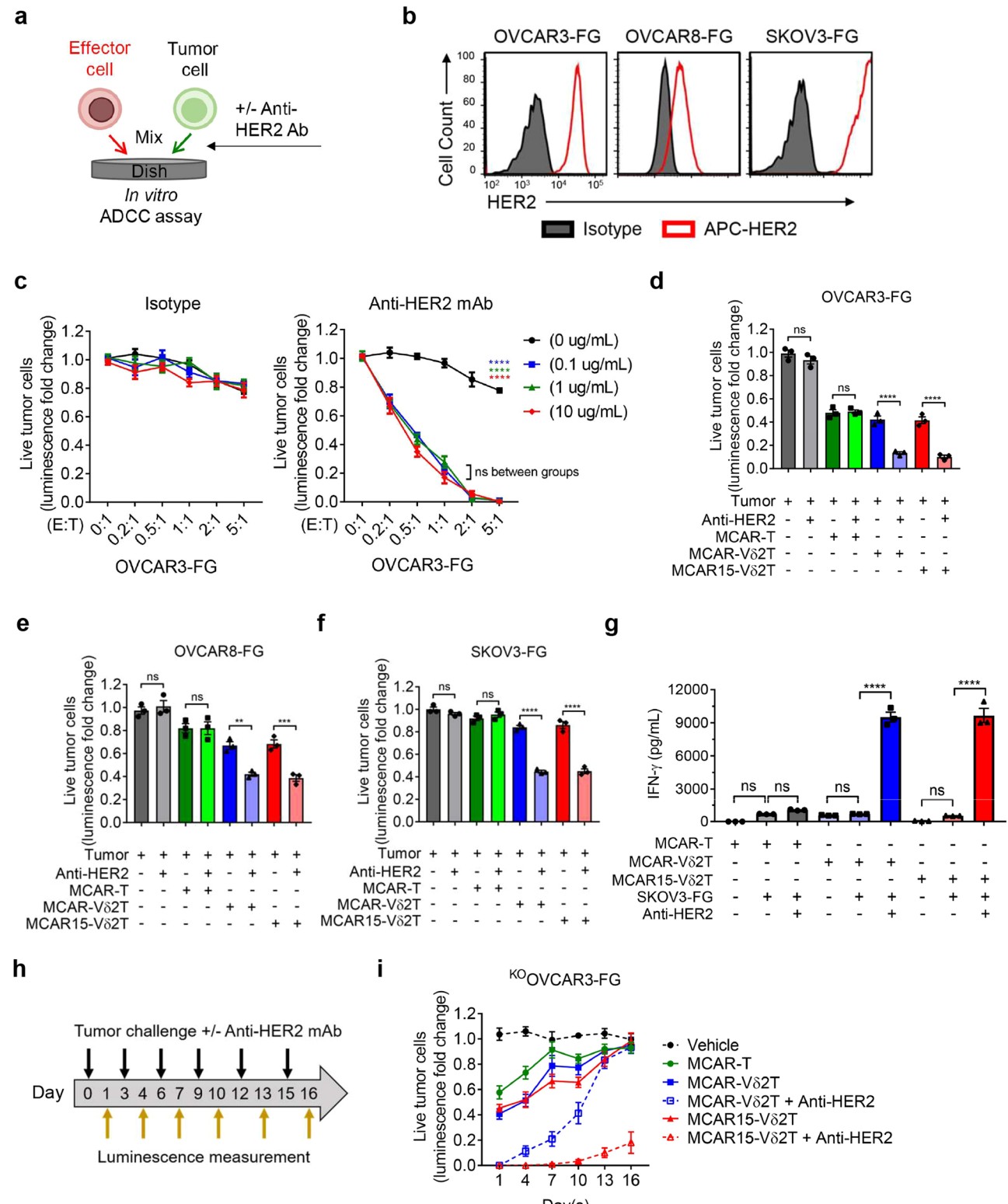

interferon gamma (IFN-γ), and tumor necrosis factor alpha (TNF-α) were purchased from PeproTech. RPMI 1640 and DMEM cell culture medium were purchased from Corning Cellgro. Fetal bovine serum (FBS) was purchased from Sigma. Medium supplements, including Penicillin/Streptomycin/Glutamine (P/S/G), MEM non-essential amino acids (NEAA), HEPES Buffer Solution, and Sodium Pyruvate, were purchased from Gibco. Beta-mercaptoethanol (β-ME) was purchased from Sigma. Normocin was purchased from InvivoGen. CryoStor cell cryopreservation media CS10 was

purchased from Sigma (Cat. C2874). Complete lymphocyte culture medium (denoted as C10 Medium) was made of RPMI 1640 supplemented with FBS (10% vol/vol), P/S/G (1% vol/vol), MEM NEAA (1% vol/vol), HEPES (10 mM), Sodium Pyruvate (1 mM), β-ME (50 mM), and Normocin (100 mg/mL). The medium for culturing OVCAR3 and OVCAR8 tumor cell line (denoted as R10 medium) was made of RPMI 1640 supplemented with FBS (10% vol/vol) and P/S/G (1% vol/vol). Medium for culturing HEK-293T/17 and SKOV3 tumor cell line (denoted as D10 medium) was made of DMEM supplemented with

**Fig. 5 | MCAR15-Vδ2T cells can also effectively target tumor cells via an antibody-dependent cell-mediated cytotoxicity (ADCC) mechanism. a–g** In vitro ADCC assay. Three human ovarian cancer cell lines (OVCAR3-FG, OVCAR8-FG, and SKOV3-FG) and four effector cells (Vδ2T, MCAR-Vδ2T, MCAR15-Vδ2T, and MCAR-T as a benchmark control) were included in the study. The same CD16^Hi donor PBMCs were used to generate all 4 types of effector cells. **a** Experimental design. Data were collected at 24 h after co-culture. Anti-HER2 Ab: monoclonal antibody trastuzumab. **b** FACS detection of HER2 expression on the indicated tumor cell lines. **c** Killing of OVCAR3-FG tumor cells by Vδ2T cells in the presence of titrated amounts of either isotype control or anti-HER2 Ab ($n = 3$). Tumor cell killing data of (**d**) OVCAR3-FG, (**e**) OVCAR8-FG, and (**f**) SKOV3-FG in the presence or absence of anti-HER2 Ab (anti-HER2 Ab concentration = 0.1 μg/mL; E:T ratio = 0.5:1; $n = 3$). **g** ELISA measurements of IFN-γ production at 24 h in the presence or absence of anti-HER2 Ab (anti-HER2 Ab concentration = 0.1 μg/mL; E:T ratio = 1:1; $n = 3$). **h, i** In vitro repeated tumor cell challenge assay studying ADCC. **h** Experimental design. Effector cells were mixed with tumor cells and rechallenged every 3 days. Tumor cell killing data was measured at 24 h post co-culture with or without the addition of anti-HER2 antibody (anti-HER2 Ab concentration = 0.1 μg/mL; E:T ratio = 2:1; $n = 3$). **i** ^KOOVCAR3-FG tumor cell killing data collected over time. Representative of 3 experiments. Data are presented as the mean ± SEM. ns, not significant; **$p < 0.01$; ***$p < 0.001$; ****$p < 0.0001$ by Student's $t$ test (**c**) or by one-way ANOVA (**d–g**). Source data and exact $p$ values are provided as a Source Data file.

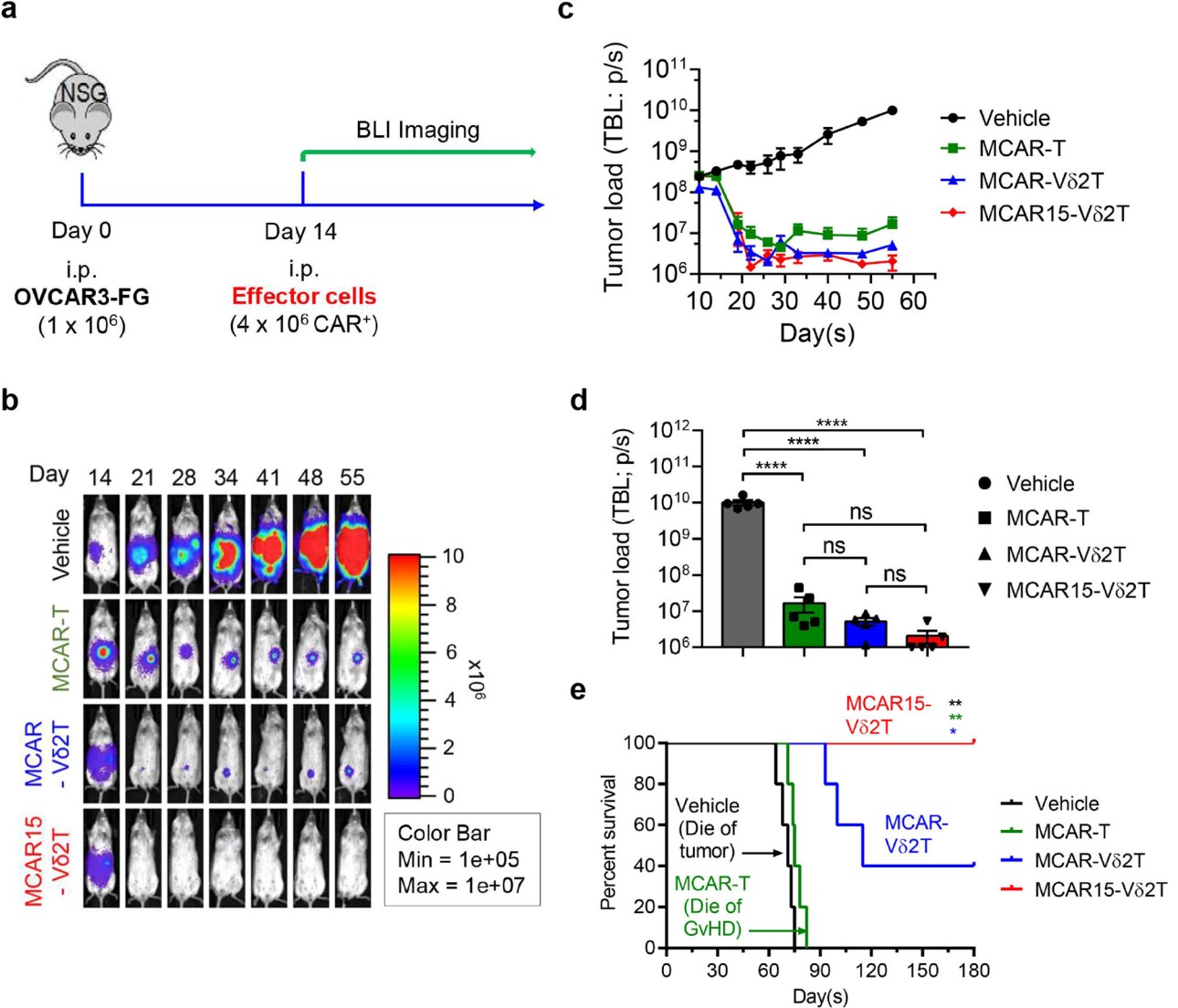

**Fig. 6 | In vivo antitumor efficacy and safety of MCAR15-Vδ2T cells in an intraperitoneal tumor model.** OVCAR3-FG human ovarian cancer cells were injected i.p. into NSG mice on Day 0, followed by i.p. administration of effector cells on day 14 ($n = 5$ mice per group). Four experimental groups were included: Vehicle (mice receiving no effector cells), MCAR-T (mice receiving MCAR-T cells), MCAR-Vδ2T (mice receiving MCAR-Vδ2T cells), and MCAR15- Vδ2T (mice receiving MCAR15-Vδ2T cells). Note, the same CD16^Hi donor PBMCs were used to generate all 3 types of effector cells. **a** Experimental design. **b** BLI images showing tumor loads in experimental mice over time. **c** Quantification of b. TBL, total body luminescence. **d** Tumor loads on day 55. **e** Kaplan–Meier survival curves of experimental mice over time. Representative of 3 experiments. Data are presented as the mean ± SEM. ns not significant; *$p < 0.05$; **$p < 0.01$; ****$p < 0.0001$ by one-way ANOVA (**d**), or by log rank (Mantel–Cox) test adjusted for multiple comparisons (**e**). Source data and exact $p$ values are provided as a Source Data file.

FBS (10% vol/vol) and P/S/G (1% vol/vol). Freezing medium for cryopreservation of cell lines and PBMC derived cells were made of CryoStor cell cryopreservation media CS10 at a 1:1 ratio with complete base medium.

## Cell lines

Human embryonic kidney 293 T/17 (HEK-293T/17, ATCC; Cat. CRL-11268), and human ovarian cancer cell lines OVCAR3 (ATCC; Cat. HTB-161), OVCAR8 (NIH; Cat. CVCL_1629), and SKOV3 (ATCC; Cat. HTB-77)

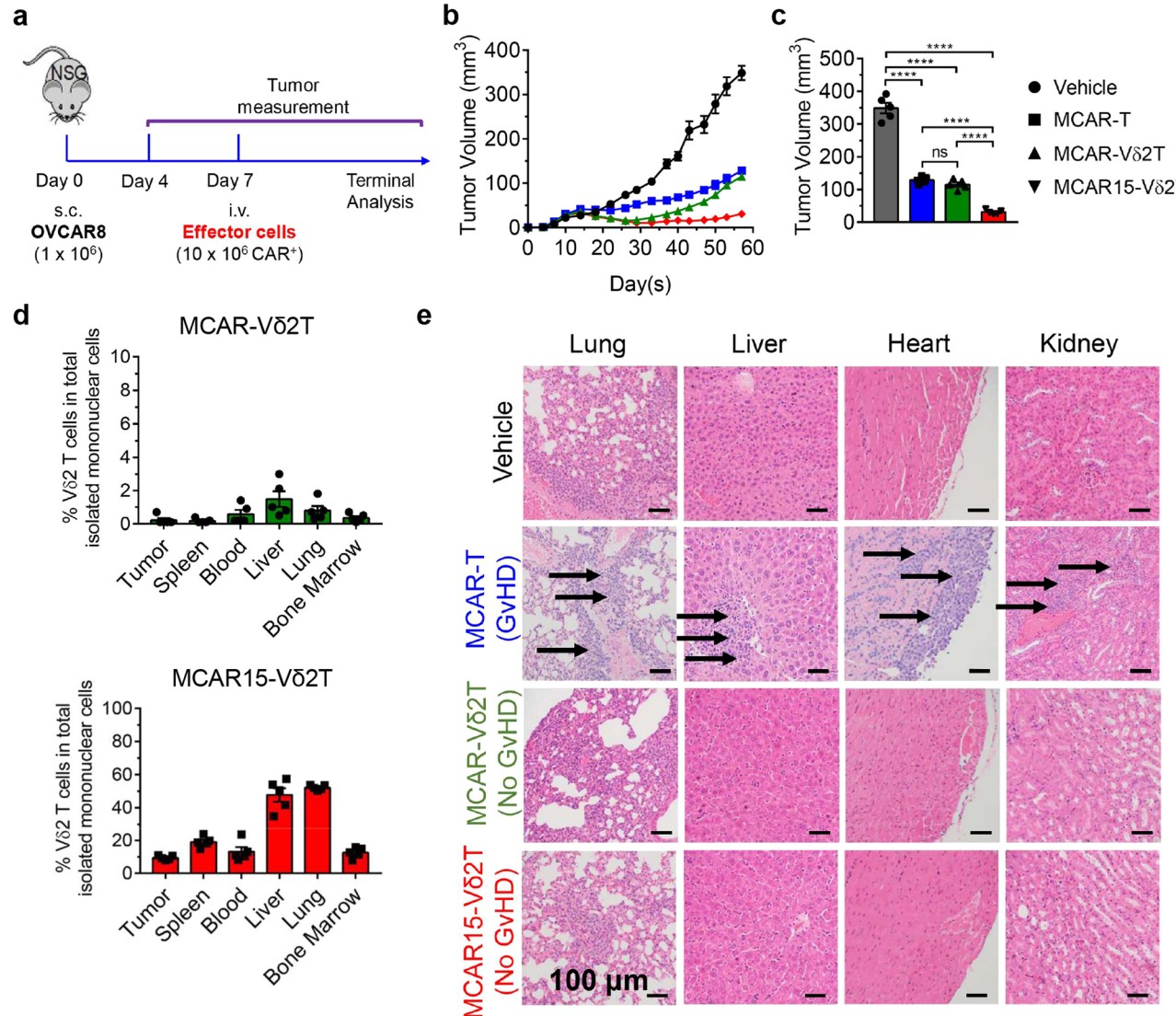

**Fig. 7 | In vivo antitumor efficacy and safety of MCAR15-Vδ2T cells in a subcutaneous tumor model.** OVCAR8 human ovarian cancer cells were injected s.c. into NSG mice on day 0, followed by intravenous (i.v.). administration of effector cells on Day 7. Four experimental groups were included: Vehicle (mice receiving no effector cells), MCAR-T (mice receiving MCAR-T cells), MCAR-Vδ2T (mice receiving MCAR-Vδ2T cells), and MCAR15-Vδ2T (mice receiving MCAR15-Vδ2T cells). Note, the same CD16^Hi donor PBMCs were used to generate all 3 types of effector cells.
**a** Experimental design. **b** Tumor growth over time. **c** Tumor size measurements collected on day 57 ($n = 5$). **d** FACS quantification of MCAR-Vδ2T and MCAR15-Vδ2T cells across tissues of experimental mice collected on day 57 ($n = 5$). **e** H&E-stained tissue sections collected on day 57. Scale bar, 100 μm. Representative of 3 experiments. Data are presented as the mean ± SEM. ns, not significant; ****$p < 0.0001$ by one-way ANOVA (**c**). Source data and exact $p$ values are provided as a Source Data file.

were all purchased from American Type Culture Collection (ATCC) or obtained from National Institutes of Health (NIH) under MTA. HEK-293T/17 and SKOV3 cell lines were maintained in D10 medium. OVCAR3 and OVCAR8 cell lines were maintained in R10 medium.

To make stable tumor cell lines overexpressing a firefly luciferase and enhanced green fluorescence protein (FG) dual reporter, the parental tumor cell lines were transduced with a Lenti/FG vector encoding the FG dual reporter[92]. 72 h-post lentivector transduction, cells were subjected to flow cytometry sorting to isolate gene-engineered cells for making stable cell lines. Three FG labeled stable tumor cell lines were generated for this study, including OVCAR3-FG, OVCAR8-FG, and SKOV3-FG.

One additional cell line that overexpresses the FG dual reporter as well as a knockout of mesothelin (MSLN) tumor antigen, ^KO^OVCAR3-FG, was generated for this study. Briefly, OVCAR3-FG cells were electroporated with a CRISPR-Cas9/MSLN-sgRNA complex composed of premixed Cas9-NLS protein (4 μL at 6.5 μg/μL; UC Berkeley) and MSLN-

sgRNA (1 μL at 100 μM; sequence: GGAAGCCAAGGAGUUGGCGA; Synthego). For electroporation, tumor cells were pulsed twice at 1170 V for 30 ms in a Neon Transfection System (Thermo Fisher Scientific; Cat. MPK5000) following manufacturer's protocol. 72 h post electroporation, the engineered OVCAR3-FG cells were subjected to flow cytometry sorting to isolate MSLN-KO OVCAR3-FG (^KO^OVCAR3-FG) cell line.

**Lentiviral vector construction**
Lentiviral vectors used in this study were all constructed from a parental lentivector pMNDW, which contains the MND retroviral LTR U2 region as an internal promoter and contains an additional truncated Woodchuck Responsive Element (WPRE) to stabilize viral mRNA[92]. The 2 A sequences derived from porcine teschovirus-1 (P2A), and Thosea asigna virus 2A (T2A) were used to link the inserted genes to achieve co-expression.

The Lenti/FG, Lenti/MCAR, Lenti/MCAR15, and Lenti/CD16 vectors were constructed by inserting into the pMNDW parental vector

corresponding the synthetic genes: a bicistronic gene encoding the FG dual reporter, a gene encoding the MSLN-targeting CAR (MCAR), a bicistronic gene encoding the same MCAR as well as a secreting form of human IL-15 (MCAR15), and a gene encoding the CD16a (CD16), respectively. The MCAR consists of SS1 scFv, CD8α hinge, CD28 transmembrane domain, CD28 signaling domain, and CD3ζ signaling domain[93]. The synthetic gene fragments were obtained from GenScript and IDT.

Lentiviruses were produced using HEK-293T/17 cells following a standard transfection protocol. Briefly, HEK-293T/17 cells were co-transfected with three plasmids: a lentiviral vector plasmid, a lentiviral glycoprotein plasmid (pCMV-VSVG), and a lentiviral packaging plasmid (pCMV-Delta R8.9), using TransIT-Lenti Transfection Reagent (Mirus Bio; Cat. MIR 6600) for 16 to 18 h. This was followed by treatment with 10 mM sodium butyrate for 8 h. Subsequently, virus-containing supernatants were generated in serum-free UltraCULTURE media (Lonza Walkersville; Cat. BP12725F) for 48 h. The supernatants were concentrated using a 100KDa Amicon Ultra-15 Centrifugal Filter Unit (Millipore Sigma; Cat. UFC910024) at 4000 rcf for 40 min at 4 °C, and stored as aliquots at −80 °C. Lentivector titers were measured by transducing HEK-293T/17 cells with serial dilutions and performing flow cytometry following established protocols.

### Antibodies and flow cytometry
Fluorochrome-conjugated antibodies specific for human APC/Cy7-CD45 (Cat. 304014, Clone H130, 1:100 dilution), PE/Cy7-TCRαβ (Cat. 306720, Clone IP26, 1:25 dilution), FITC-CD3 (Cat. 317306, Clone OKT3, 1:200 dilution), PB-CD3 (Cat. 317314, Clone OKT3, 1:100 dilution), FITC-CD27 (Cat. 356403, Clone M-T271, 1:100 dilution), APC/Cy7-CD45RA (Clone HI100, 1:200 dilution), FITC-CD56 (Cat. 318304, Clone HCD56, 1:10 dilution), APC-TCR Vγ9 (Cat. 331309, Clone B3, 1:100 dilution), PerCP-TCR Vδ2 (Cat. 331410, Clone B6, 1:100 dilution), APC-CD16 (Clone 3G8, 1:250 dilution), PE-CCR2 (Cat. 357206, Clone K036C2, 1:400 dilution), PE/Cy7-CCR4 (Cat. 359410, Clone L291H4, 1:500 dilution), FITC-CCR5 (Cat. 359120, Clone J418F1, 1:200 dilution), PE/Cy7-CXCR3 (Cat. 353720, Clone G025H7, 1:100 dilution), APC/Cy7-IFN-γ (Cat. 502529, Clone B27, 1:100 dilution), FITC-granzyme B (Cat. 372205, Clone QA16A02, 1:1000 dilution), PE/Cy7-perforin (Cat. 308125, Clone dG9, 1:50 dilution), PE-pSTAT5 (Tyr694, Cat. 936903, Clone A17016B, 1:50 dilution), PE/Cy7-Bcl-2 (Cat. 633511, Clone BCL/10C4, 1:100 dilution), APC-HER2 (Cat. 324407, Clone 24D2, 1:400 dilution), PB-CD14 (Cat. 301815, Clone 63D3, 1:1000 dilution), FITC-CD11b (Cat. 301330, Clone ICRF44, 1:10000 dilution), APC/Cy7-CD163 (Cat. 333622, Clone GHI/61, 1:500 dilution), APC-CD206 (Cat. 321110, Clone 15-2, 1:500 dilution), and APC-Streptavidin (Cat. 405207, 1:1000 dilution) to bind Biotinylated Human Mesothelin for MCAR staining were all purchased from BioLegend.

Biotinylated Human Mesothelin (Cat. MSN-H82E9, 1:400 dilution) was purchased from ACROBiosystems. Fluorochrome-conjugated antibody specific for human APC-Mesothelin (Cat. FAB32652A, Clone 420411, 1:100 dilution) was purchased from R&D Systems. Fluorochrome-conjugated antibody specific for human FITC-Bcl-xL (Cat. MA5-28637, Clone 7B2.5, 1:100 dilution) was purchased from Thermo Fisher Scientific. Fluorochrome-conjugated antibody specific for human FITC-TCRγ/δ (Cat. 347903, Clone 11F2, 1:15 dilution) was purchased from BD Biosciences. Fluorochrome-conjugated antibody specific for human PE-TCR Vδ1 (Cat. 130-120-580, Clone REA173, 1:500 dilution) was purchased from Miltenyi Biotec. Fixable Viability Dye eFluor506 (e506, Cat. 65-0866-18) was purchased from Affymetrix eBioscience. Mouse Fc Block (anti-mouse CD16/32, Cat. 553142, Clone 2.4G2, 1:50 dilution) was purchased from BD Biosciences, and human Fc Receptor Blocking Solution (TrueStain FcX, Cat. 422302, 1:25 dilution) was purchased from BioLegend. InVivoSIM anti-human HER2 (Trastuzumab Biosimilar, Cat. SIM0005) was purchased from BioXCell.

Flow cytometry surface stainings and intracellular stainings were performed following standard protocols, as well as specific instructions provided by a manufacturer for particular antibodies. Intracellular staining of IL15-mediated pro-survival signaling pathway molecules (pSTAT5, Bcl-2, and Bcl-xL) were performed following Foxp3/Transcription Factor Staining protocol (Thermo Fisher Scientific; Cat. 50-112-8857). Stained cells were analyzed using a MACSQuant Analyzer 10 flow cytometer (Miltenyi Biotech). FlowJo software version 10 (BD Biosciences) was used for data analysis.

### Enzyme-Linked Immunosorbent Cytokine Assays (ELISA)
The ELISA for detecting human IFN-γ was performed following a standard protocol from BD Biosciences. Supernatants from cell culture assays were collected and assayed to quantify IFN-γ. The capture (Cat. 551221, Clone NIB42, 1:250 dilution) and biotinylated (Cat. 554550, Clone 4 S.B3, 1:500 dilution) pairs for detecting IFN-γ were purchased from BD Biosciences. The HRP-Avidin conjugate (Cat. 405103, 1:1000 dilution) and the human IFN-γ ELISA standards (Cat. 570209) were purchased from BioLegend. 1-Step™ TMB ELISA Substrate Solutions was purchased from Thermo Fisher Scientific (Cat. 34021). Human IL-15 was quantified with Human IL-15 Quantikine ELISA Kit (R&D Systems; Cat. D1500). The samples were analyzed for absorbance at 450 nm using an Infinite M1000 microplate reader (Tecan).

### Generation of PBMC-derived conventional αβ T cells and derivatives
Healthy donor human PBMCs were obtained from the UCLA/CFAR Virology Core Laboratory, with identification information removed under federal and state regulations. Protocols using these human cells were exempted by the UCLA Institutional Review Board (IRB), IRB #05-10-093, 21 January 2019. To generate PBMC-derived conventional αβ T (denoted as PBMC-T) cells, $1 \times 10^6$ cells/mL PBMCs were resuspended in C10 medium supplemented with 100 IU/mL human IL-2 (T-medium) and stimulated with 50 ng/mL of the anti-CD3 monoclonal antibody OKT3 (BioLegend; Cat. 317325). 2 days after activation, PBMCs were washed and passaged 3 times per week for 2 weeks to maintain a cell density at $0.5–1 \times 10^6$ cells/mL; fresh T-medium was added at every passage.

To generate MCAR-T cells, $1 \times 10^6$ cells/mL PBMCs were stimulated with 50 ng/mL of the anti-CD3 monoclonal antibody OKT3 in the T-medium. 2 days after activation of the PBMC cultures, cells were washed, resuspended in the fresh T-medium, and then concentrated MCAR lentivector was added to the PBMC cultures. The following day, transduced cells were washed and passaged 3 times per week for 2 weeks to maintain a cell density at $0.5 - 1 \times 10^6$ cells/mL; fresh T-medium was added at every passage. The resulting MCAR-T cells were collected and cryopreserved for future use.

### Generation of PBMC-derived Vδ2 T cells and derivatives
Healthy donor human PBMCs were obtained from the UCLA/CFAR Virology Core Laboratory, with identification information removed under federal and state regulations. Protocols using these human cells were exempted by the UCLA Institutional Review Board (IRB), IRB #05-10-093, 21 January 2019. To generate PBMC-derived Vδ2 T (denoted as Vδ2T) cells, $2.5 \times 10^6$/mL PBMCs were resuspended in C10 medium supplemented with 100 IU/mL human IL-2 (T-medium) and stimulated with 5 μM ZOL. 3 days after the activation, Vδ2T cells were enriched via TCRγ/δ+ T Cell Isolation Kit (Miltenyi Biotech; Cat. 130-092-892), and then resuspended in the fresh T-medium. PBMCs were washed and passaged 3 times per week for 10–14 days to maintain a cell density at $1–1.5 \times 10^6$ cells/mL; fresh T-medium was added at every passage.

To generate MCAR-Vδ2T and MCAR15-Vδ2T cells, $2.5 \times 10^6$/mL PBMCs were resuspended in the T-medium and stimulated with 5 μM ZOL. 3 days after activation, PBMCs were washed, enriched via TCRγ/δ+ T Cell Isolation Kit, resuspended in the fresh T-medium, and then

concentrated MCAR or MCAR15 lentivector was added to the PBMC cultures. The following day, transduced cells were washed and passaged 3 times per week for 2 weeks to maintain a cell density at $1 - 1.5 \times 10^6$ cells/mL; fresh T-medium was added at every passage. The resulting MCAR-Vδ2T and MCAR15-Vδ2T cells were collected and cryopreserved for future use.

## RNA-Seq analysis of Vδ2T cells

A total of 13 PBMC-derived Vδ2T cell samples were analyzed. Vδ2T cells were expanded and purified according to Fig. 1a. Cell samples were sorted using a FACSAria II flow cytometer (BD Biosciences). Total RNAs were isolated from each cell sample using a miRNeasy Mini Kit (QIAGEN). cDNAs were synthesized using an iScript cDNA Synthesis Kit (BioRad). Libraries were constructed using an Illumina TruSeq Stranded Total RNA Sample Prep kit and sequenced with 50 bp single-end reads (targeting $20 \times 10^6$ reads per sample) on Illumina HiSeq3000 following the manufacturer's instructions and the TCGB Core's standard protocol. The raw sequencing data underwent quality check using FastQC software (version 0.11.9), and fastp (version 0.23.2) was employed to remove sequencing adapters and low-quality bases. The trimmed reads were mapped to the human reference genome (hg38) with STAR 2.7.9a and the gene count matrixes were obtained using featureCounts from Subread package (version 2.0.3). Batch effect removal was carried out using Combat-Seq, which was implemented in the sva package (version 3.44.0). Gene expression counts normalized by sequencing depth (cpm, counts per million) were obtained using edgeR (version 3.38.4). A log-transformation with pseudocount 1 was applied for principal component analysis (PCA) and heatmap generation. The top 500 most variable genes were used for PCA plots. Differential expression analyses were carried out using DESeq2 (version 1.36.0) based on CD16 expression. Gene with exactly 0 $p$-value was represented as the smallest $p$-value divide by 10. To control for false discovery rate (FDR), the Benjamini–Hochberg (BH) procedure was applied to adjust $p$-values, and genes with an adjusted $p$-value below the 0.05 threshold were identified as differentially expressed. Gene set enrichment analysis was performed based on all genes' differential analysis results using clusterProfiler (version 4.4.4). Gene Ontology over-representation analysis was performed on the differential expressed genes and a Cnet plot was used to display the linkages between differentially expressed genes and the enriched biological processes. The immune cell-type score of each Vδ2 T cell samples were calculated using ssGSEA method based on gene signatures of the 24 immune cell-type from ImmuneCellAI, and the correlations between enrichment scores of the designated immune cell type and the samples' CD16 expression level were plotted using R package corrplot (version 0.92).

## In vitro Vδ2T cell phenotype and function analyses

The phenotype of Vδ2T cells and derivatives were studied using flow cytometry, by analyzing cell surface markers including MCAR and MCAR15 expression, memory T cell markers (i.e., CD27 and CD45RA), chemokine receptors (i.e., CXCR3, CCR2, CCR4, and CCR5), and NK receptors (i.e., CD56). The capacity of these cells to produce cytotoxic molecules (i.e., perforin and granzyme B) was studied using flow cytometry via intracellular staining. Intracellular staining of IL15-mediated pro-survival signaling pathway molecules (pSTAT5, Bcl-2, and Bcl-xL) were performed following Foxp3/Transcription Factor Staining protocol. The proliferation of Vδ2T cells was measured by cell counting and flow cytometry (identified as CD3$^+$TCR Vδ2$^+$) over time.

## In vitro 24 h tumor cell killing assay

FG-labeled tumor cells ($1 \times 10^4$ cells per well) were co-cultured with effector cells (at indicated ratios) in Corning 96-well clear bottom black plates for 24 h, in C10 medium with or without the addition of ZOL (5 μM). At the end of the culture, live tumor cells were quantified

by adding D-luciferin (150 μg/mL; Caliper Life Science) to cell cultures and reading out luciferase activities using an Infinite M1000 microplate reader (Tecan).

## In vitro repeated tumor challenge assay

The main Figs. 4f and 5h illustrate the experimental design employed in this study. Briefly, FG-labeled tumor cells ($1 \times 10^4$ cells per well) were co-cultured with effector cells (at indicated ratios) in six Corning 96-well clear bottom black plates containing C10 medium with or without the addition of ZOL (5 μM) or anti-HER2 Ab (0.1 μg/mL). 24 h later, live tumor cells from one plate (1st time point) were quantified by adding D-luciferin to cell cultures and reading out luciferase activities using an Infinite M1000 microplate reader. On day 3, the remaining five plates were centrifuged at 300 rcf for 5 min, and the old medium was carefully replaced with fresh C10 medium. Subsequently, the cells from these five plates were resuspended and transferred to five newly seeded tumor cell plates. After 24 h, live tumor cells from one plate (2nd time point) were quantified by measuring luciferase activities. This process was repeated for a total of six time points, spanning 16 days.

## In vitro ADCC assay

FG-labeled tumor cells ($1 \times 10^4$ cells per well) were seeded and subjected to treatment with anti-HER2 Ab (BioXCell; InVivoSIM anti-human HER2; Cat. SIM0005) at 37 °C for 30 min in Corning 96-well clear bottom black plates containing C10 medium. Subsequently, effector cells were added to the antibody treated tumor cells and co-cultured them for 24 h. At the end of the culture, live tumor cells were quantified by adding D-luciferin to cell cultures and reading out luciferase activities using an Infinite M1000 microplate reader (Tecan).

## In vitro M2-Polarized Macrophage killing assay

Healthy donor human PBMCs were obtained from the UCLA/CFAR Virology Core Laboratory, with identification information removed under federal and state regulations. Protocols using these human cells were exempted by the UCLA Institutional Review Board (IRB), IRB #05-10-093, 21 January 2019. PBMCs were cultured in serum-free RPMI 1640 media (Corning cellgro, Manassas, VA, USA, #10-040-CV) at $1 \times 10^7$ cells/mL cell density. Subsequently, 10–15 mL of the PBMC suspension was seeded into a 10 cm dish and incubated for 1–2 h in a humidified 37 °C, 5% CO2 incubator. Next, the medium containing non-adherent cells was discarded and the dishes were washed twice using PBS. The adherent monocytes were cultured in C10 medium, and human M-CSF (10 ng/mL; PeproTech; Cat. 300-25) for 6 days to generate monocyte-derived macrophages (MDMs). At day 6, the generated MDMs were dissociated by 0.25% Trypsin/EDTA (Gibco; Cat. 25200-056), collected, and reseeded in a 6-well plate in C10 medium at 0.5–1 × 10$^6$ cells/mL for 48 h in the presence of recombinant human IL-4 (10 ng/mL; PeproTech; Cat. 214-14) and human IL-13 (10 ng/mL; PeproTech; Cat. 214-13) to induce MDM polarization. Polarized MDMs were then collected and used for flow cytometry or for setting up in vitro mixed culture experiments.

## In vivo bioluminescence live animal imaging (BLI)

BLI was performed using a Spectral Advanced Molecular Imaging (AMI) HTX imaging system (Spectral instrument Imaging). Live animal imaging was acquired 5 min after intraperitoneal (i.p.) injection of D-Luciferin (1 mg/mouse) for total body bioluminescence. Imaging results were analyzed using the AURA imaging software (Spectral Instrument Imaging).

## In vivo antitumor efficacy study in an OVCAR3 human ovarian cancer xenograft NSG mouse model

(i.p. tumor inoculation mimicking orthotopic growth of ovarian cancer). The experimental design is shown in the main Fig. 6a. Briefly, on day 0, NSG mice received intraperitoneal (i.p.) inoculation of OVCAR3-

FG cells ($1 \times 10^6$ cells per mouse). On day 14, the experimental mice were assayed for tumor burden using BLI and then placed into 4 equivalent BLI-expressing groups. On the same day, the experimental mice either received i.p. injection of vehicle (PBS) or effector cells ($4 \times 10^6$ CAR$^+$ cells/mouse in PBS). All mice were monitored for survival and their tumor loads were measured twice per week using BLI.

## In vivo antitumor efficacy study in an OVCAR8 human ovarian cancer xenograft NSG mouse model

(s.c. tumor inoculation mimicking solid tumor growth of ovarian cancer). The experimental design is shown in the main Fig. 7a. Briefly, on day 0, NSG mice received subcutaneous (s.c.) inoculation of OVCAR8 cells ($1 \times 10^6$ cells per mouse). On day 7, the experimental mice were assayed for tumor burden and then placed into 4 equivalent tumor size groups. On the same day, the experimental mice either received intravenous (i.v.) injection of vehicle (PBS) or effector cells ($10 \times 10^6$ CAR$^+$ cells/mouse in PBS). All mice were monitored for survival and their tumor loads were measured twice per week using a Fisherbrand™ Traceable™ digital caliper (Thermo Fisher Scientific). Tumor volume was calculated using the formula: Volume (mm$^3$) = (length × width$^2$)/2. At the end of the experiments, mice were terminated. Solid tumors were retrieved, weighted using a PA84 precision balance (Ohaus), then processed for flow cytometry analysis to detect tumor-infiltrating Vδ2 T cells (identified as hCD45$^+$Vδ2$^+$ cells). Various mouse tissues (blood, heart, lung, liver, and kidney) were also harvested and processed for flow cytometry analysis to detect tissue biodistribution of the Vδ2 T cells, following established protocols[94].

## In vivo antitumor efficacy study to compare $^{16H}$MCAR15-Vδ2T and $^{16L}$MCAR15-Vδ2T cells

The experimental design is shown in the Supplementary Fig. 5a. Briefly, on day 0, NSG mice received subcutaneous (s.c.) inoculation of OVCAR8 cells ($1 \times 10^6$ cells per mouse). On day 14, the experimental mice were assayed for tumor burden and then placed into 4 equivalent tumor size groups. On the same day, the experimental mice either received intravenous (i.v.) injection of vehicle (PBS) or effector cells ($5 \times 10^6$ CAR$^+$ cells/mouse in PBS). All mice were monitored for survival and their tumor loads were measured twice per week using calipers. Tumor volume was calculated using the formula: Volume (mm$^3$) = (length × width$^2$)/2.

## Histopathologic analysis

Tissues (i.e., spleen, lung, liver, heart, and kidney) were collected from the experimental mice and fixed in 10% neutral buffered formalin for up to 36 h and embedded in paraffin for sectioning (5 μm thickness). Tissue sections were prepared and stained with Hematoxylin and Eosin (H&E) by the UCLA Translational Pathology Core Laboratory, following the Core's standard protocols. Stained sections were imaged using an Olympus BX51 upright microscope equipped with an Optronics Macrofire CCD camera (AU Optronics) at 20 x and 40 x magnifications. The images were analyzed using Optronics PictureFrame software (AU Optronics).

## Statistics

Rstudio and Graphpad Prism 7 software (Graphpad) were used for statistical data analysis. Student's two-tailed $t$ test was used for pairwise comparisons. Ordinary one-way ANOVA followed by Tukey's or Dunnett's multiple comparisons test was used for multiple comparisons. Log rank (Mantel−Cox) test adjusted for multiple comparisons was used for Meier survival curves analysis. Data are presented as the mean ± standard error of the mean (SEM), unless otherwise indicated. In all figures and figure legends, "n" represents the number of samples or animals utilized in the indicated experiments. A $P$ value of less than 0.05 was considered significant. ns not significant; *$p < 0.05$; **$p < 0.01$; ***$p < 0.001$; ****$p < 0.0001$.

## Reporting summary

Further information on research design is available in the Nature Portfolio Reporting Summary linked to this article.

## Data availability

The bulk RNAseq data sets generated in this study have been deposited in the GEO database under accession code GSE235755. The remaining data associated with this study are presented in the Article or Supplementary Information. Further information and requests may be directed to and will be fulfilled by the corresponding author, Lili Yang (liliyang@ucla.edu). Source data are provided with this paper.

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

## Acknowledgements

We thank the University of California, Los Angeles (UCLA) animal facility for providing animal support; the UCLA Translational Pathology Core Laboratory (TPCL) for providing histology support; the UCLA Technology Center for Genomics & Bioinformatics (TCGB) facility for providing RNA-Seq services; the UCLA CFAR Virology Core for providing human PBMCs; the UCLA BSCRC Flow Cytometry Core Facility for cell sorting support; and the UCLA QCBio Collaboratory for the bioinformatics analysis support. We also want to thank BioRender.com. Figures 1a and 3a, Supplementary Fig. 4a, d were created with BioRender.com under UCLA institutional license (Agreement number: HP25YA3ZU3). This work was supported by a UCLA Faculty Startup Fund (to L.Y.), a UCLA BSCRC Innovation Award (to L.Y.), and an Ablon Scholars Award (to L.Y.). D.L. was a predoctoral fellow supported by a T32 Microbial Pathogenesis Training Grant (Ruth L. Kirschstein National Research Service Award, T32-AI007323), and a postdoctoral fellow supported by a T32 Tumor Immunology Training Grant (USHHS Ruth L. Kirschstein Institutional National Research Service Award, T32-CA009120). Z.L. was a postdoctoral fellow supported by a T32 Tumor Immunology Training Grant (USHHS Ruth L. Kirschstein Institutional National Research Service Award, T32-CA009120). Y.-R.L. was a postdoctoral fellow supported by a UCLA MIMG M. John Pickett Postdoctoral Fellow Award and a UCLA BSCRC Postdoctoral Fellowship. J.Z. was supported by NIH (R21HL150374, R01HG006139) and National Science Foundation (DMS-2054253, IIS-2205441).

## Author contributions

D.L., Z.S.D., W.G., P.W., and L.Y. designed the study, analyzed the data, and wrote the manuscript. D.L. performed all experiments, with assistance from Z.S.D., W.G., C.J.R., N.E.P., Y.Y., K.Z., Z.L., F.M., M.L., T-C.S., X.C., and Y.-R.L. Y.Y., and K.Z. generated the MSLN knockout OVCAR3-FG cell line. Y.Y. generated human M2-polarized macrophages and

performed the in vitro M2 Mφ killing assay. W.G. analyzed RNA-Seq data. J.Z. helped with the statistical analysis of data. M.P., P.W., and L.Y. supervised the entire study.

## Competing interests

D.L., Z.S.D., and L.Y. are inventors on patents (UC Case No. 2023-043, Title: Enhanced Gamma Delta T Cells for Immunotherapy) related to this work filed by UCLA. Currently, Z.S.D. is employed at Guggenheim Securities, N.E.P. is employed at RTM Law, APC, Z.L. is employed at Allogene, F.M. is employed at Amgen, and Y.Y. is employed at Retro Biosciences. P.W. is a co-founder, stockholder, consultant, and advisory board member of Simnova Bio, TCRCure Biopharma, and Appia Bio. L.Y. is a scientific advisor to AlzChem and Amberstone Biosciences, and a co-founder, stockholder, and advisory board member of Appia Bio. None of the declared companies contributed to or directed any of the research reported in this article. The remaining authors declare no competing interests.
