## [Peer Review File · Nature Communications]

Unlocking the potential of allogeneic V δ 2 T cells for ovarian cancer therapy through CD16 biomarker selection and CAR/IL-15 engineeringEditorial Note: Parts of this Peer Review File have been redacted as indicated to remove third-party material where no permission to publish could be obtained.

REVIEWER COMMENTS

Reviewer #1 (Remarks to the Author): with expertise in V δ 2 T cells, immuno-engineering, cancer immunology

Review of Unlocking the Potential of Allogeneic V δ 2 T Cells for Ovarian Cancer Therapy through CD16 Biomarker Selection and CAR/IL-15 Engineering

The paper provides several conceptual advances, the most important of which is the in vivo experiments in figures 6 and 7 which show that V δ 2 cells engineered to express secreted IL-15 give longer disease control than cells lacking IL-15 and do this in the absence of xenogeneic graft versus host disease. The data are demonstrated with two different CARs in three different target lines and two different animal models.

Whilst there is some additional gain of function using the CAR, it is noticeable that in contrast with conventional alpha beta T cells, the CAR-g δ T are capable of killing antigen negative cancer cell targets within the tumour. This is an important conceptual advantage over conventional CAR-T approaches against tumours with antigen expression heterogeneity.

Minor issues

Refs 19 and 20 should be the twin seminal papers originally describing BTN3A1 and 2A1 heterodimers recognising v γ 9v δ 2. Hence the paper in immunity 2020 from Willcox should be cited as ref 19 and not the subsequent paper from Lai

Figure 1H. it is interesting that there is no cytokine , perforin or granzyme response in the absence of ZOL; some cancer targets will engage V δ 2 cells without needing zol pre-treatment. It is worth a sentence in the text to point out the zol dependency in this

particular target line.

Figure 3i is interesting but a bit confusing. There is a very much larger IL-15 production by the MCAR15 cells in the presence of antigen than in its absence following 48 cultures in the presence of the target cells. Indeed in the absence of target the MCAR15 cells seem to produce no more IL-15 than non-transduced even though IL-15 is driven by a strong promoter. Is this because of CAR-T cell expansion in the presence of antigen over the 48 hour period or is it caused by CAR-T cell death during this time period in the absence of target cell?

There is no mention of any cytokine being used during manufacture apart from the legend to Fig 3a which states that IL-15 stands for interleukin 15, even though that symbol does not appear in the schematic. Could you clarify if any cytokine were used in manufacture and that the results in 3i cannot be explained by cytokine withdrawal?

Fig 3j, although many people in the field accept the canonical designation of memory markers based on RA and 27 equally applies to gamma delta T cells, this is an area of contention and it is worth pointing out that the existence for example of TEMRA in gdT is somewhat controversial

Figure 5. the text mentions "ADCC is mediated by CD16 receptor, which presents in two forms – ". There should be reference however to the high affinity Fcγ receptors CD32 and CD64 which are arguably more important for ADCC via macrophages, and also expressed although less brightly on gamma delta T cells

Supplemental figure 4. Very interesting experiment; the MCAR15 are shown to be cytotoxic to M2 macrophages even in the absence of ZOL. It would be interesting to know if the IL-15 component is needed for the macrophage killing.

Figure 6 shows impressive survival in the absence of GvHD especially in the CAR co-expressing IL-15. It would be very informative to know if the engineered T cells persisted in peritoneum and if the long term survival in the IL-15 group led to disease free animals. I understand the tissues might not be available from these animals for reasons of technical difficulty and would not necessarily be ethically warranted to repeat the experiment in

order to gain additional PD data.

Reviewer #2 (Remarks to the Author): with expertise in $\gamma\delta$ T cells, adoptive cell therapy

This manuscript describes the development and characterization of CAR and IL-15 engineered CD16Hi V δ 2 T cells for the development of allogeneic cellular immunotherapies. The authors utilized a simple method for producing V δ 2 T cells with superior antitumor activity by screening donors for CD16 expression on V δ 2 T cells. Further, the authors utilized mesothelin-targeted CAR and IL-15 engineering to enhance the antitumor efficacy of CD16Hi V δ 2 T cells and demonstrated the manufacturability, cancer therapy potential, and high safety profile of engineered CD16Hi V δ 2 T cells. The experimental design was reasonable, and the methods were detailed and reproducible, supporting the authors' hypothesis.

To improve the manuscript, I would suggest the following points:

1. It would be helpful to provide clinical characteristics for healthy donors, such as age, sex, or other health conditions, which could affect the characteristics of V δ 2 T cells.
2. The authors suggested possible mechanisms of enhanced cytotoxicity of CD16Hi V δ 2 T cells based on gene signatures. However, it would be interesting to know if cytotoxicity of V δ 2 T cells varied linearly if donors were further divided into groups according to CD16 expression.
3. It is recommended that the authors explore ADCC through CD16 using other antibodies in further research.
4. The authors should discuss the differences between using IL-5 combined with CAR T cells and IL-5 transduced CAR T cells.
5. The authors should provide a possible explanation for why MSLN knockout resulted in reduced killing by CAR-V δ 2 T cells in the absence of ZOL compared to with ZOL after 24 h, whereas no difference was seen in the parental OVCAR3-FG cell line.

6. It would be helpful for the authors to explain whether the cytotoxicity of V δ 2 T cells is largely dependent on ADCC since the isotype control had minimal impact on V δ 2 T cell cytotoxicity, whereas significant enhancement of tumor killing was seen with anti-HER2 mAb.

7. It would be beneficial for the authors to provide experimental data on the superior durability of MCAR15 V δ 2 T cells since they stated that MCAR15 V δ 2 T cells were more durable, resulting in sustained remission.

8. The authors should address the possible limitation of the penetration of CAR T cells into the tumor in cellular therapy in solid tumors and provide any data on the presence of CAR T cells inside the tumor.

9. Finally, it would be helpful to include a discussion of potential limitations, challenges, or adverse effects in the manuscript as this is an essential aspect to consider in the development of new therapies, despite the extensive discussion of the potential advantages and benefits of CD16Hi V δ 2 T cells as a cancer therapy.

Reviewer #3 (Remarks to the Author): with expertise in CAR-T, ovarian cancer

In Lee et al manuscript, the authors develop a gdT (V δ 2) cell strategy engineered with CAR and IL-15 to improve therapeutic responses against ovarian cancer. They first evaluated how CD16 expression impacts function of V δ 2 T cells, and further engineered V δ 2 cells with CAR and IL-15 to evaluate efficacy and safety in xenograft models of ovarian cancer. This is an interesting study that sheds light on impact of CD16 as a biomarker for V δ 2 cells, and the potential for CAR-IL15 engineering of CD16hi V δ 2 cells for ovarian cancer therapy. Several questions should be addressed to raise the impact of this study:

1. What is the impact of Zol on the in vitro functionality of CD16hi and lo V δ 2 cells? Data should be included in 1g to support tumor cell killing ability with and without Zol.
2. Does the addition of Zol negate the need to select for CD16hi cells?
3. Does Zol impact CD16 expression directly?

4. How does Zol impact transcriptional signatures in CD16hi and lo cells, this should be evaluated in Figure 2, since Zol has been implicated in Th17 signatures. This would be important to look at in the context of these studies.
5. It's unclear the importance of Fig 1i and j. If enriching CD16hi cells improved cytotoxicity, but engineering CD16 into CD16lo has little effects on function, what is the purpose of showing this data beyond showing "CD16 is a biomarker" with little direct impact on cell function?
6. It is unclear whether the remainder of these studies used donors with CD16hi Vd2 cells? Please clarify.
7. Does IL-15 engineering impact memory phenotype of CARs? Fig 3j and k only include CAR+ and – cells, but not CAR+IL15+ and CAR+IL15- cells.
8. The controls (MCAR-Vd2 and MCAR15-Vd2 cells) for Fig 4g and 5i show significant variability in killing ability between the two experiments. Are these experimental designs the same? Please explain.
9. Does the ADCC benefit occur in vivo? Some study in vivo would be beneficial.
10. Is CAR activity required for therapeutic effects in Fig 6? Based on Fig 4, likely the CAR is not required for therapy.
11. What is the persistence of T cells in blood and ascites in this model following i.p. T cell administration? The response differences and GVHD should be evaluated further.
12. It is likely that increased levels of CAR-IL15 Vd2 cells in non-tumor tissue would have deleterious effects over time. Please explain.
13. Does this version of IL15 cross react to mouse cells? If so, please clarify or reference literature, which would support the use of these xenograft models to evaluate safety of this approach and specifically of engineered IL15.
14. Do CAR-IL15 Vd2 cells also kill M1 macrophages? What is the mechanism of action? Do CAR-IL15 Vd2 cells kill M2 macrophages better than CAR Vd2 cells?

Reviewer #4 (Remarks to the Author): with expertise in ovarian cancer, cancer immunology

General comments:

This study by Lee and Dunn et al. investigated the potential of allogeneic Vgamma9Vdelta2

(Vd2) T cells as a therapeutic cell product in adoptive cell therapies for ovarian cancers. Authors first discovered that baseline CD16 expression on Vd2 is highly variable between individuals and Vd2 T-cell products from CD16Hi donors show better cytotoxicity and cytokine production. In addition, authors successfully demonstrated that CD16Hi Vd2 cells expressing mesothelin-targeting CAR show potent anti-tumor effects, which is further potentiated by co-expression of IL-15, against ovarian cancer cell lines both in vitro and in vivo. In contrast to conventional alpha/beta T cells, Vd2 CAR T cells caused no xenogeneic GvHD in mice. Finally, it was demonstrated that CD16Hi, but not CD16Lo, Vd2 T cells show ADCC against ovarian cancer cell lines using a HER2-specific antibody. Together, this study supports the use of allogeneic CD16Hi Vd2 T cells in cell therapies for cancers.

Major comments:

1. One of strengths of this study is the identification of a subset of individuals who have CD16Hi Vd2 T cells as donors of therapeutic Vd2 T cells. However, in contrast to detailed comparison between CD16Hi and CD16Lo donors, CD16Hi and CD16Lo Vd2 T cells within a CD16Hi or CD16Lo donor were not characterized. It would be important to investigate phenotype, proliferation, and anti-tumor functions of CD16Hi and CD16Lo Vd2 T cells within a donor.
2. In all experiments testing gamma/delta TCR-mediated anti-tumor effects, ZOL was used to sensitize cancer cells to Vd2 T cells. Significance of TCR-mediated anti-tumor effects of Vd2 T cells in the presence of ZOL should be explained or discussed.
3. Cytotoxicity of Vd2 CAR T cells against M2-polarized macrophages is an attractive and unique attribute of Vd2 T cells in comparison to other immune effector cells. However, only MCAR15-engineered Vd2 T cells were tested in these experiments. It would be important to clarify roles of CAR and IL-15 in the cytotoxicity against M2 macrophages. In addition, it is important to test cytotoxicity against other myeloid cells such as M1/M0 macrophages, monocytes, and dendritic cells.

Minor comments:

1. In figure 3i, it appears that production of IL-15 requires co-culture with cancer cells, which is unlikely because IL-15 is co-expressed with CAR by a constitutive promoter. Is it possible that transgenic expression of IL-15 enhances production of endogenous IL-15 production

from Vd2 T cells?

2. For experiments of repeated stimulations (for example figure 4f-g), it is unclear whether the T cells are harvested and counted after each stimulation, so that T cell/cancer cell ratio is constant in all stimulations.
3. For supplementary fig 1, NCR2 was described in the text, but not appeared in the Figure.
4. Figure 4i was not explained in the text.
5. The source of mesothelin specific CAR or scFv and detailed information for “preclinical anti-HER2 IgG1 analogue to trastuzumab” should be provided.
6. Although the manuscript is generally well-written and straightforward, “methods” section should be carefully revised. It appears that many reagents and procedures in this section were not used in the experiments in this study. Electroporation/Nucleofection condition of “250 V for 5 minutes” is impossible. Finally, a little more detail of lentiviral vector production and transduction should be provided: for example, packaging/envelope plasmids, timing of harvests, and a use of transduction enhancers.

Point-to-point responses

Reviewer #1 (Remarks to the Author): with expertise in Vδ2 T cells, immuno-engineering, cancer immunology

Review of Unlocking the Potential of Allogeneic Vδ2 T Cells for Ovarian Cancer Therapy through CD16 Biomarker Selection and CAR/IL-15 Engineering

The paper provides several conceptual advances, the most important of which is the in vivo experiments in figures 6 and 7 which show that Vδ2 cells engineered to express secreted IL-15 give longer disease control than cells lacking IL-15 and do this in the absence of xenogeneic graft versus host disease. The data are demonstrated with two different CARs in three different target lines and two different animal models.

Whilst there is some additional gain of function using the CAR, it is noticeable that in contrast with conventional alpha beta T cells, the CAR-gdT are capable of killing antigen negative cancer cell targets within the tumour. This is an important conceptual advantage over conventional CAR-T approaches against tumours with antigen expression heterogeneity.

Response: Thank you for your comments and critiques on our paper. We appreciate your thoughtful analysis and are pleased to hear that you found the in vivo experiments in Figures 6 and 7 to be significant conceptual advances. We agree that these findings are important, as they demonstrate the effectiveness of Vδ2 cells engineered to express secreted IL-15 in providing longer disease control than cells lacking IL-15 without causing xenogeneic graft versus host disease, as well as the potential advantages of using CAR-gdT cells over conventional CAR-T approaches against tumors with antigen expression heterogeneity. We have incorporated your feedback into our paper, and we feel these improvements enhance the strength of our paper. Once again, we thank you for your insightful comments on our paper.

Minor issues

1.1 Refs 19 and 20 should be the twin seminal papers originally describing BTN3A1 and 2A1 heterodimers recognising Vg9vd2. Hence the paper in Immunity 2020 from Willcox should be cited as ref 19 and not the subsequent paper from Lai.

Response 1.1: We appreciate the Reviewer's comment. Following the Reviewer's suggestion, we have updated Refs 19 and 20 to be the twin seminal papers originally describing BTN3A1 and 2A1 heterodimer recognition by Vδ2 T cells.

1.2 Figure 1H. It is interesting that there is no cytokine, perforin or granzyme response in the absence of ZOL; some cancer targets will engage Vδ2 cells without needing ZOL pre-treatment. It is worth a sentence in the text to point out the ZOL dependency in this particular target line.

Response 1.2: Following the Reviewer's suggestion, we have included a sentence highlighting the ZOL dependency of this target line (as well as the others used in our studies) and that some cancer targets will engage Vδ2 cells without needing ZOL pre-treatment. We have also included new references to address this topic. We have included the following: "Depending

on the cancer cell type and assay used, V δ 2 T cells are capable of tumor killing and producing cytokine, perforin, and granzyme responses in the absence of ZOL; in some cases, ZOL or other preconditioning is used exert effective cancer killing^{31,32}. However, for the ovarian cancer cells we tested, V δ 2 T cells exhibited a dependency on ZOL for cytotoxicity and effector molecule production during in vitro cocultures (Fig. 1h).”

1.3 Figure 3i is interesting but a bit confusing. There is a very much larger IL-15 production by the MCAR15 cells in the presence of antigen than in its absence following 48 cultures in the presence of the target cells. Indeed in the absence of target the MCAR15 cells seem to produce no more IL-15 than non-transduced even though IL-15 is driven by a strong promoter. Is this because of CAR-T cell expansion in the presence of antigen over the 48 hour period or is it caused by CAR-T cell death during this time period in the absence of target cell?

Response 1.3: We have added a statement and sources to support the consistency of our findings. We have included “IL-15 secretion was validated by ELISA (Fig. i), and these findings are consistent with previously published data that demonstrate increased IL-15 production by CAR/sIL-15 engineered $\alpha\beta$ T⁴³, V δ 1 T⁴⁴, and NK cells²³ following antigen stimulation. This may be due to the heightened metabolic activity and protein translation that occur during cell activation as well as the short half-life of IL-15.”

1.4 There is no mention of any cytokine being used during manufacture apart from the legend to Fig 3a which states that IL-15 stands for interleukin 15, even though that symbol does not appear in the schematic. Could you clarify if any cytokine were used in manufacture and that the results in 3i cannot be explained by cytokine withdrawal?

Response 1.4: We appreciate the Reviewer’s comment. Only IL-2 and ZOL were used to activate and expand V δ 2 T cells, and this clarification has been added to the figure legend (Fig. 3a legend). IL-15 was not used in manufacturing, and thus the results in 3i cannot be explained by cytokine withdraw. We have also moved the IL-15 abbreviation definition to Fig 3b. We thank the reviewer for helping us enhance the clarity of Fig 3.

1.5 Fig 3j, although many people in the field accept the canonical designation of memory markers based on RA and 27 equally applies to gamma delta T cells, this is an area of contention and it is worth pointing out that the existence for example of TEMRA in gdT is somewhat controversial

Response 1.5: We have included a statement to acknowledge the differing views on V δ 2 T cell memory status. We have included the following: “Although various definitions of V δ 2 T cell memory status do exist in literature^{44,45}, our analysis based on CD27 and CD45RA expression showed that both CAR+ and CAR- populations from the MCAR15-V δ 2 T cell group were mostly central memory (~ 40%) and effector memory (~ 50%) phenotypes (Fig. 3j-l).”

1.6 Figure 5. the text mentions “ADCC is mediated by CD16 receptor, which presents in two forms – “. There should be reference however to the high affinity Fc γ receptors CD32 and CD64 which are arguably more important for ADCC via macrophages, and also expressed although less brightly on gamma delta T cells

Response 1.6: We have revised our paragraph on ADCC and included a statement and sources that address the potential role of CD32 and CD64. We have added the following: "T and NK cell-mediated ADCC is predominantly attributed to the CD16a (FcγRIIIa) transmembrane receptor, which is expressed by many effector cells of the immune system, whereas CD16b (FcγRIIIb), a GPI-anchored protein, is exclusively expressed by neutrophils⁴⁸. Although CD32 (FcγRI) and CD64 (FcγRII) are expressed at low levels on Vδ2 T cells and may contribute to ADCC, they are primarily implemented in myeloid-mediated ADCC^{49,50}. We thus focus our Vδ2 T cell studies on the canonical lymphocyte CD16a receptor, which we refer to as CD16."

1.7 Supplemental figure 4. Very interesting experiment; the MCAR15 are shown to be cytotoxic to M2 macrophages even in the absence of ZOL. It would be interesting to know if the IL-15 component is needed for the macrophage killing.

Response 1.7: We appreciate the Reviewer's comment. We have previously performed a similar experiment using unmodified Vδ2 T cells (no CAR and IL-15 engineering). The data was published on (*Cancers* **2022**, 14(11), 2749; <https://doi.org/10.3390/cancers14112749>). As shown in Figure 5 of that paper, Vδ2 T cells were capable of killing M2 macrophages without CAR and IL-15 engineering. With the addition of ZOL, the killing was further enhanced. This suggests that CAR and IL-15 not required for M2 macrophage killing, but may augment the killing (i.e. potentially due to increased persistence with IL-15 expression).

[Editorial Note: Figure redacted]

Figure 5 from *Cancers* **2022**, 14(11), 2749, showing the killing of M2-polarized macrophages by Vδ2 T cells. (A) Experimental design. Zoledronate was added to activate γδT cells. (B) FACS analysis of live macrophages 24 h after co-culturing with γδT cells. (C) ELISA analysis of IFN-γ secretion by γδT cells in the supernatants of various mixed cell cultures (n = 3). (D) FACS detection of CD25 expression on γδT cells. (E) Quantification of D (n = 3). Representative of three experiments.

1.8 Figure 6 shows impressive survival in the absence of GvHD especially in the CAR co-expressing IL-15. It would be very informative to know if the engineered T cells persisted in peritoneum and if the long term survival in the IL-15 group led to disease free animals. I understand the tissues might not be available from these animals for reasons of technical

difficulty and would not necessarily be ethically warranted to repeat the experiment in order to gain additional PD data.

Response 1.8: We thank the Reviewer for the critique of our study. We appreciate your suggestion to provide more information on the persistence of engineered T cells in the peritoneum and the disease-free survival of animals in the IL-15 group and agree that these data would have been valuable additions to our study. We also appreciate your understanding that obtaining tissues for such analyses may not always be feasible due to technical and ethical considerations. However, we will certainly take this feedback into account for future studies and strive to provide as much information as possible within the limits of what is practical and ethical. Although a different model, we want to highlight the persistency and immunohistochemistry data shown in Fig 7 as evidence that V δ 2 T cells persist and do not cause xenoreactivity in vivo.

Reviewer #2 (Remarks to the Author): with expertise in $\gamma\delta$ T cells, adoptive cell therapy

This manuscript describes the development and characterization of CAR and IL-15 engineered CD16Hi V δ 2 T cells for the development of allogeneic cellular immunotherapies. The authors utilized a simple method for producing V δ 2 T cells with superior antitumor activity by screening donors for CD16 expression on V δ 2 T cells. Further, the authors utilized mesothelin-targeted CAR and IL-15 engineering to enhance the antitumor efficacy of CD16Hi V δ 2 T cells and demonstrated the manufacturability, cancer therapy potential, and high safety profile of engineered CD16Hi V δ 2 T cells. The experimental design was reasonable, and the methods were detailed and reproducible, supporting the authors' hypothesis.

Response: Thank you for taking the time to review our manuscript and for your valuable comments and critiques. We are delighted to hear that you found our work to be a significant contribution to the development of allogeneic cellular immunotherapies. We appreciate your recognition of our use of a simple method for producing CD16Hi V δ 2 T cells with superior antitumor activity, as well as our utilization of mesothelin-targeted CAR and IL-15 engineering to enhance the antitumor efficacy of these cells and your comments on the experimental design and the reproducibility of our methods. We have integrated your feedback into our paper, and we believe that these changes have strengthened the quality and impact of our work. Thank you again for taking the time to review our paper, we appreciate your insightful comments and contributions to our research.

To improve the manuscript, I would suggest the following points:

2.1 It would be helpful to provide clinical characteristics for healthy donors, such as age, sex, or other health conditions, which could affect the characteristics of V δ 2 T cells.

Response 2.1: We appreciate your suggestion to provide more information on donor characteristic and agree that these data would have been valuable additions to our study. Unfortunately, we checked with our donor sources, and we do not have access to these characteristics for the donors used in this study. However, we will certainly take this feedback

into account for future studies and strive to provide as much information as possible within the limits of what is practical and ethical.

2.2 The authors suggested possible mechanisms of enhanced cytotoxicity of CD16^{Hi} Vδ2 T cells based on gene signatures. However, it would be interesting to know if cytotoxicity of Vdelta2 T cells varied linearly if donors were further divided into groups according to CD16 expression.

Response 2.2: We appreciate the Reviewer's comment. This is a beneficial analysis to conduct and following the Reviewer's suggestion, we have compared the tumor killing activity based on CD16 expression, as shown in **Figure R1**. Our data indicates that the cytotoxicity of Vδ2 T cells vary linearly if donors were further divided into groups according to CD16 expression. These results indicate that for future developments, identifying and collecting PBMCs from donors with the highest CD16 expression may be beneficial for creating Vδ2 T cell products with the most potent antitumor killing.

2.3 It is recommended that the authors explore ADCC through CD16 using other antibodies in further research.

Response 2.3: We have conducted additional ADCC studies using anti-CD20 mAb for targeting Raji. As shown in Figure R2, CD16^{Hi} Vδ2 T cells exert ADCC when cultured with Raji cells and anti-CD20 mAb. Since our manuscript is focused on ovarian cancer, we decided not to include this in the manuscript.

2.4 The authors should discuss the differences between using IL-5 combined with CAR T cells and IL-5 transduced CAR T cells.

Response 2.4: IL-15 self-secretion strategy, in comparison to combination therapy with IL-15 injections, potentially allows for the sustained, local delivery of IL-15 to the CAR-T cell environment. This can encourage the delivery of IL-15 to the transduced CAR-T cells, simplify treatment regimens, and reduce systemic toxicity. We have included a statement in the manuscript as well as a new source that highlights the potential benefits of IL-15 secretion compared to IL-15 injections. We have included the following in our discussion: "Although IL-15 injections can increase circulating NK and CD8+ T cells, achieving sustained IL-15 signaling using soluble IL-15 is difficult due to its short serum half-life and limited bioavailability⁷⁶. IL-15 self-secretion has the potential to provide sustained and local delivery of IL-15 to engineered immune cells, as well as simplify treatment regimens and reduce systemic toxicity."

2.5 The authors should provide a possible explanation for why MSLN knockout resulted in reduced killing by CAR-Vδ2 T cells in the absence of ZOL compared to with ZOL after 24 h, whereas no difference was seen in the parental OVCAR3-FG cell line.

Response 2.5: We appreciate the Reviewer's comment. The manuscript describes Fig 4i as the bar graph showing the killing data from our readout 24hr after the third tumor rechallenge, in accordance with the Fig 4 legend. The parental line OVCAR3-FG has the CAR target mesothelin expressed, which allows targeting by the MCAR. As shown in Fig 4c, the addition

of ZOL has little effect on the CAR-V δ 2 T cell killing of OVCAR3-FG cells, whereas against MLSN KO OVCAR3-FG, ZOL is required for pronounced killing by CAR-V δ 2 T cells. This supports the conclusion that CAR-V δ 2 T cells target OVCAR3-FG cancer cells through CAR and TCR recognition (upon the addition of ZOL), whereas MCAR-T cells are not able to kill MLSN KO OVCAR3-FG cells. We have added a clarifying statement in our manuscript. "MSLN knockout resulted in reduced killing by CAR-V δ 2 T cells in the absence of ZOL compared to with ZOL after the repeated tumor challenge (Fig. 4i), whereas no difference was seen in cytotoxicity towards the parental OVCAR3-FG cell line after 24 h cocultures (Fig. 4c). This indicates that in the absence of ZOL, the killing of parental OVCAR3-FG is driven by CAR-mediated killing."

2.6 It would be helpful for the authors to explain whether the cytotoxicity of V δ 2 T cells is largely dependent on ADCC since the isotype control had minimal impact on V δ 2 T cell cytotoxicity, whereas significant enhancement of tumor killing was seen with anti-HER2 mAb.

Response 2.6: We appreciate the Reviewer's comment. We have revised the figure legend and manuscript to properly reflect the study of **unmodified CD16^{Hi} V δ 2 T cells** in Fig 5c. In Fig 5d-i, MCAR and MCAR15 engineered CD16^{Hi} V δ 2 T cells were studied. We believe it is useful to show the ADCC effect for both unmodified and engineered CD16^{Hi} V δ 2 T cells. As shown in Fig 1c, unmodified CD16^{Hi} V δ 2 T cells have minimal killing of OVCAR3-FG cells in the absence of ZOL. Thus, it is likely that ADCC due to the addition of anti-HER2 mAb results in the enhanced killing observed in Fig 5c. We thank the reviewer for drawing this to our attention and have corrected our manuscript and legend.

2.7 It would be beneficial for the authors to provide experimental data on the superior durability of MCAR15 V δ 2 T cells since they stated that MCAR15 V δ 2 T cells were more durable, resulting in sustained remission.

Response 2.7: We agree that more information on MCAR15 V δ 2 T cells in the intraperitoneal model would have been valuable additions to our study. We appreciate your understanding that obtaining additional tissues for such analyses may not always be feasible due to technical and ethical considerations. In Fig 6, we shown that it is only the MCAR15-V δ 2 T cell-treated group that has 5/5 survival out to 180 days, indicating durable/long-term tumor control, whereas 3 mice in the MCAR-V δ 2 T cells died due to tumor growth. This provides in vivo support that IL-15 increases durable tumor control. We have altered our phrasing to "MCAR15-V δ 2T cells resulted in complete remissions in all 5/5 mice through day 180 without any signs of GvHD, highlighting the benefit of IL-15 engineering in long-term intraperitoneal tumor growth inhibition (Fig. 6e)" instead of durability to clarify that our statement is regarding tumor control, rather than cell therapy persistence. Although a different model, we want to highlight the persistency and immunohistochemistry data shown in Fig 7 as evidence that MCAR15-V δ 2 T cells persist and do not cause xenoreactivity in vivo. We also believe our in vitro repeated tumor challenge assays in Fig 4g and Fig 5i also provide experimental data on the long-term functionality of MCAR15-V δ 2 T cells.

2.8 The authors should address the possible limitation of the penetration of CAR T cells into the tumor in cellular therapy in solid tumors and provide any data on the presence of CAR T cells inside the tumor.

Response 2.8: We appreciate the Reviewer's comment. Tumor penetration is an important consideration in developing CAR T cells to treat solid tumors. We kindly point to Fig 7d as preliminary evidence that MCAR15-Vδ2 T cells infiltrate subcutaneous tumors in vivo, given the significant increase in MCAR15-Vδ2 T cells present in the tumor compared to MCAR-Vδ2 T cells at the time of terminal harvest. We have added a statement in our manuscript to highlight the possible limitation of the penetration of CAR T cells into solid tumors: "Tumor penetration is an important consideration in developing CAR T cells to treat solid tumors, and while there is preliminary evidence that MCAR15-Vδ2T cells infiltrate subcutaneous tumors in vivo, additional studies and ultimately clinical investigation will be needed to show meaningful tumor penetration in patients. Upon infiltrating the tumor, sufficient persistence and antitumor functionality are significant clinical challenges, as immunorejection and the immunosuppressive tumor microenvironment may thwart CD16^{Hi} Vδ2 T cell-based therapies."

2.9 Finally, it would be helpful to include a discussion of potential limitations, challenges, or adverse effects in the manuscript as this is an essential aspect to consider in the development of new therapies, despite the extensive discussion of the potential advantages and benefits of CD16^{Hi} Vδ2 T cells as a cancer therapy.

Response 2.9: We agree that is essential to discuss the potential limitations and challenges of new therapies. We highlighted areas for further development, such as cell memory status, in vivo polarization, immunogenicity, and CD16^{Hi} Vδ2 T cell pool formation, in which we include challenges (i.e. host-mediated rejection), and now we have added an additional section in our discussion to clarify limitations faced by engineered CD16^{Hi} Vδ2 T cell-based therapies. We appreciate the Reviewer's guidance to dedicate a separate paragraph to recognize limitations.

Reviewer #3 (Remarks to the Author): with expertise in CAR-T, ovarian cancer

In Lee et al manuscript, the authors develop a gdT (Vd2) cell strategy engineered with CAR and IL-15 to improve therapeutic responses against ovarian cancer. They first evaluated how CD16 expression impacts function of Vd2 T cells, and further engineered Vd2 cells with CAR and IL-15 to evaluate efficacy and safety in xenograft models of ovarian cancer. This is an interesting study that sheds light on impact of CD16 as a biomarker for Vd2 cells, and the potential for CAR-IL15 engineering of CD16^{hi} Vd2 cells for ovarian cancer therapy. Several questions should be addressed to raise the impact of this study:

Response: Thank you for taking the time to review our manuscript and providing us with your valuable comments and critiques. We are excited to hear that you consider our work to be an interesting study for the development of Vd2 cells and engineered Vd2 cells for ovarian cancer therapy. We have carefully incorporated your feedback into our paper, and we believe that your input has enhanced the quality and impact of our research. Thank you again for your valuable feedback and for your time spent reviewing our paper.

3.1 What is the impact of Zol on the in vitro functionality of CD16^{hi} and lo Vd2 cells? Data should be included in 1g to support tumor cell killing ability with and without ZOL.

Response 3.1: Although V δ 2 T cells are capable of producing cytokine, perforin, and granzyme responses in the absence of ZOL when engaging certain cancer targets, V δ 2 T cells exhibited a dependency on ZOL for cytotoxicity and effector molecule secretion during in vitro co-cultures with OVCAR3-FG. Without the addition of ZOL, killing of OVCAR3-FG was not observed. We have added the without ZOL groups to Fig 1g, and we kindly point to Fig 1h showing that CD16^{Hi} V δ 2 T and CD16^{Lo} V δ 2 T cells produce limited amounts of IFN γ , Perforin, and Granzyme B in the absence of ZOL when cultured with OVCAR3-FG cells. We have added a statement to our manuscript highlighting the ZOL dependency of the target line OVCAR3-FG.

3.2 Does the addition of Zol negate the need to select for CD16hi cells?

Response 3.2: We appreciate the Reviewer's comment. As shown in Fig 1g, ZOL is added to both CD16^{Hi} V δ 2 T and CD16^{Lo} V δ 2 T cells and the killing of OVCAR3-FG and SKOV3-FG cancer cells are assessed. CD16^{Hi} V δ 2 T cells exhibit significantly enhanced cytotoxicity compared to CD16^{Lo} V δ 2 T cells, indicating that ZOL does not negate the need to select for CD16^{Hi} V δ 2 T cells.

3.3 Does Zol impact CD16 expression directly?

Response 3.3: We appreciate the Reviewer's comment. As shown in Fig 1c-d, CD16 expression on CD16^{Hi} V δ 2 T cells was not only maintained but increased upon V δ 2 T cell activation with ZOL and expansion for 14 days. For CD16^{Hi} and CD16^{Lo} donors, CD16 expression stays the same or increases. Importantly, CD16^{Hi} donors maintain higher expression of CD16 post ZOL expansion compared to CD16^{Lo} V δ 2 T cells, indicating that the pre-expansion classification of CD16^{Hi} or CD16^{Lo} results in post-expansion cells that remain in the same category.

3.4 How does Zol impact transcriptional signatures in CD16hi and lo cells, this should be evaluated in Figure 2, since Zol has been implicated in Th17 signatures. This would be important to look at in the context of these studies.

Response 3.4: We are assessing CD16^{Hi} and CD16^{Lo} V δ 2 T cells post-expansion to characterize V δ 2 T cells for the development of donor-derived V δ 2 T cell-based therapies. Our focus is on differential gene expression based on CD16 post-expansion, and all donor cells were expanded using the same zoledronate stimulation method. We understand that the Reviewer suggests investigating the transcriptional information of V δ 2 T cells with and without zoledronate for future studies. However, since our cells require zoledronate for expansion, we did not investigate without zoledronate. Nonetheless, we agree that this would be an interesting area of exploration for future V δ 2 T cell studies, although it may need to be conducted with unexpanded cells unless non-zoledronate means of expansion are used. We appreciate the reviewer's insights and feedback.

3.5 It's unclear the importance of Fig 1i and j. If enriching CD16hi cells improved cytotoxicity, but engineering CD16 into CD16lo has little effects on function, what is the purpose of showing this data beyond showing "CD16 is a biomarker" with little direct impact on cell function?

Response 3.5: We appreciate the Reviewer's comment. As the reviewer has recognized, the purpose of Fig 1i-k (previously Fig 1i-j) are to confirm that CD16 expression is a biomarker for donor selection for expanding Vδ2 T cells with increased cytotoxicity function. Although CD16 is well-studied and not known to have direct cytotoxic functions, we believe it is useful to shown that transgenic expression of CD16 does not increase the cytotoxicity of CD16^{Lo} Vδ2 T cells, and thus genetic engineering of CD16^{Lo} Vδ2 T cells to express CD16 will not recapitulate the enhanced cytotoxicity of CD16^{Hi} Vδ2 T cells. We have added a statement to our manuscript to highlight this consideration: "This suggests that CD16 could be used as a biomarker to select for donors with highly potent Vδ2 T cells rather than functioning as an active receptor that enhances tumor killing, and that genetic introduction of CD16 to CD16^{Lo} Vδ2 T cells may not recapitulate the heightened activity of Vδ2 T cells expanded from the CD16^{Hi} donors."

3.6 It is unclear whether the remainder of these studies used donors with CD16^{hi} Vδ2 cells? Please clarify.

Response 3.6: Yes, all subsequent studies are performed using CD16^{Hi} Vδ2 T cells unless otherwise stated. We have updated our manuscript and figures to make this clarification.

3.7 Does IL-15 engineering impact memory phenotype of CARs? Fig 3j and k only include CAR+ and – cells, but not CAR+IL15+ and CAR+IL15- cells.

Response 3.7: We appreciate the Reviewer's comment. No, IL-15 engineering does not impact memory phenotype of CARs. For Fig 3k and 3l, we show memory phenotype of CAR15+ and CAR15- cells within MCAR15-Vδ2 T cells. Because we used bicistronic lentiviral vector (MCAR-IL15) to engineer our cells, so the cells would express CAR and IL-15 together (we relabeled figures as CAR15+/- rather than CAR+/- to make it more clear). In Figure R3, we also have included the data of memory phenotype of CAR+ and CAR- cells (without IL-15 engineering) below. As shown, IL-15 engineering does not alter the memory phenotype of the CAR-engineered Vδ2 T cells after expansion.

3.8 The controls (MCAR-Vd2 and MCAR15-Vd2 cells) for Fig 4g and 5i show significant variability in killing ability between the two experiments. Are these experimental designs the same? Please explain.

Response 3.8: We appreciate the Reviewer's comment. The experimental designs are similar with several key differences. For Fig 4g, OVCAR3-FG (left) and OVCAR8-FG (right) cells are used as the target tumor cell lines, and the repeated tumor challenge assay is run with and without the addition of ZOL. For Fig 5i, MSLN KO OVCAR3-FG cells are used as the target tumor cell line, and the assay is run with and without the additional of anti-HER2 mAb. We believe these differences, including different target cells and ZOL or mAb supplement, account for the significantly variable killing between the two experiments. Against MSLN KO OVCAR3-FG cells used in Fig 5i, the addition of anti-HER2 mAb enables significant killing of the tumor target cells, whereas in Fig 4g OVCAR3-FG are efficiently killed by effector cells with and without zoledronate due to CAR-mediated killing.

3.9 Does the ADCC benefit occur in vivo? Some study in vivo would be beneficial.

Response 3.9: Based on the previous studies in the literature such as ADCC for NK cells, in vitro ADCC activity tends to translate well into the in vivo benefit. We plan to perform studies to investigate the in vivo ADCC function and will publish the result in a separate manuscript.

3.10 Is CAR activity required for therapeutic effects in Fig 6? Based on Fig 4, likely the CAR is not required for therapy.

Response 3.10: We appreciate the Reviewer's comment. Non-engineered Vδ2 T cells do not kill OVCAR3-FG in the absence of ZOL, whereas CAR-engineered Vd2 T cells kill OVCAR3-FG in

the absence of ZOL. This indicates that CAR is needed to kill OVCAR3-FG cells in the absence of ZOL. In Fig 4, all the effector cells are expressing CAR, and the data support the benefit of expressing CAR on the effector cells. During the in vivo study in Fig 6, ZOL is not administered, and thus CAR-engineered cells are explored for assessing in vivo antitumor efficacy.

3.11 What is the persistence of T cells in blood and ascites in this model following i.p. T cell administration? The response differences and GVHD should be evaluated further.

Response 3.11: We appreciate your suggestion to provide more information on the persistence of engineered T cells in the peritoneum and the disease-free survival of animals in the IL-15 group and agree that these data would have been valuable additions to our study. We also appreciate your understanding that obtaining tissues for such analyses may not always be feasible due to technical and ethical considerations. However, we will certainly take this feedback into account for future studies and strive to provide as much information as possible within the limits of what is practical and ethical. Although a different model, we want to highlight the persistency and immunohistochemistry data shown in Fig 7 as evidence that Vδ2 T cells persist and do not cause xenoreactivity in vivo.

3.12 It is likely that increased levels of CAR-IL15 Vδ2 cells in non-tumor tissue would have deleterious effects over time. Please explain.

Response 3.12: We appreciate the Reviewer's comment. As shown in Fig 7, CAR15-Vδ2 T cells persist in non-tumor tissue without causing deleterious effects. In contrast, the immunohistochemistry data shows that MCAR-T result in the accumulation of mononuclear cells, indicative of preclinical GvHD. This indicates that although the CAR15-Vδ2 T are present, there is no observed alloreactivity of the Vδ2 TCR and minimal risk of GvHD. Normal mesothelin expression is highly restricted to the mesothelium, and mesothelin-targeting CARs have been proven safe in the clinic to date as well as Vδ2 T cell therapies, indicating that the persistence of mesothelin-targeting CAR15-Vδ2 T cells would be unlikely to cause damage to essential tissue.

3.13 Does this version of IL15 cross react to mouse cells? If so, please clarify or reference literature, which would support the use of these xenograft models to evaluate safety of this approach and specifically of engineered IL15.

Response 3.13: We appreciate the Reviewer's comment. Yes, this version of IL15 cross reacts to mouse cells. We believe it is common to use NSG mice to study IL-15 immune cell effector functions, and we have included several references that support the use of these xenograft models to evaluate the safety of this approach and specifically of engineered IL15. In the three studies, xenograft NSG tumor models are used to study the safety and efficacy of soluble IL-15 expressing, CAR engineered immune cells. For two of the three studies, clinical trials were ultimately performed using analogous soluble IL-15 expressing cell products.

Reference 1: Batra et al. "Glypican-3-Specific CAR T Cells Coexpressing IL15 and IL21 Have Superior Expansion and Antitumor Activity against Hepatocellular Carcinoma." *Clinical Cancer Research*, 2020: Transgenic expression of IL15 and/or IL21 for enhancing glypican-3-CAR (GPC3-CAR) T cells' antitumor properties against HCC was explored. Cytokine expressing

GPC3-CAR T-cell antitumor activity in murine intraperitoneal xenograft models of GPC3+ tumors was assessed. This is a preclinical therapy analogous to the engineered IL-15 expressing CAR-T cell product used in the clinical trial NCT04377932.

Reference 2: Makkouk, A. et al. "Off-the-shelf V δ 1 gamma delta T cells engineered with glypican-3 (GPC-3)-specific chimeric antigen receptor (CAR) and soluble IL-15 display robust antitumor efficacy against hepatocellular carcinoma." *J. Immunother. Cancer* 9, (2021): In a subcutaneous HepG2 mouse model in immunodeficient NSG mice, GPC-3.CAR/sIL-15 V δ 1 T cells primarily accumulated and proliferated in the tumor, and a single dose efficiently controlled tumor growth without evidence of xenogeneic GvHD. Importantly, compared with GPC-3.CAR V δ 1 T cells lacking sIL-15, GPC-3.CAR/sIL-15 V δ 1 T cells displayed greater proliferation and resulted in enhanced therapeutic activity.

Reference 3: Liu et al. "Cord blood NK cells engineered to express IL-15 and a CD19-targeted CAR show long-term persistence and potent antitumor activity." *Leukemia*, 2018: CB-derived NK cells were transduced with a retroviral vector incorporating the genes for CAR-CD19, IL-15 and inducible caspase-9-based suicide gene (iC9), and demonstrated efficient killing of CD19-expressing cell lines and primary leukemia cells in vitro, with marked prolongation of survival in a xenograft Raji lymphoma murine model. Interleukin-15 (IL-15) production by the transduced CB-NK cells critically improved their function. This is a preclinical therapy analogous to the engineered CB-NK cell product used in the clinical trial NCT03056339.

3.14 Do CAR-IL15 V δ 2 cells also kill M1 macrophages? What is the mechanism of action? Do CAR-IL15 V δ 2 cells kill M2 macrophages better than CAR V δ 2 cells?

Response 3.14: We appreciate the Reviewer's comment. We previously performed a similar experiment using unmodified V δ 2 T cells (no CAR and IL-15 engineering). The data was published in *Cancers* **2022**, 14(11), 2749; <https://doi.org/10.3390/cancers14112749>. As shown in Figure 5 of the previous publication, V δ 2 T cells were capable of killing M2 macrophages without CAR and IL-15 engineering. With the addition of ZOL, the killing was further enhanced. Our previous results and our new data in Supplementary Fig 4 indicate that CAR and IL-15 are not required for M2 macrophage killing. These assays were performed using the same 24 h coculture, and the killing with and without engineering is comparable. For both engineered and nonengineered V δ 2 T cells, ZOL increases M2 macrophage killing, which implicates the V δ 2 TCR in macrophage killing. In our studies, we focus on M2 macrophages, and there is literature reporting that both M1 and M2 macrophages were susceptible to V δ 2 T cell cytotoxicity with the addition of ZOL (*Cancer Immunol Immunother* 66, 1205–1215 (2017). <https://doi.org/10.1007/s00262-017-2011-1>). The authors conclude that that ZOL can render M1 and M2 M ϕ s susceptible to V δ 2+ T cell cytotoxicity in a perforin-dependent manner. The mechanism of action is likely through the recognition of V δ 2 TCR with BTN3A1/2A1 expressed by macrophages, although more studies are needed to validate this mechanism.

Reviewer #4 (Remarks to the Author): with expertise in ovarian cancer, cancer immunology

General comments:

This study by Lee and Dunn et al. investigated the potential of allogeneic Vgamma9Vdelta2 (Vd2) T cells as a therapeutic cell product in adoptive cell therapies for ovarian cancers. Authors first discovered that baseline CD16 expression on Vd2 is highly variable between individuals and Vd2 T-cell products from CD16^{Hi} donors show better cytotoxicity and cytokine production. In addition, authors successfully demonstrated that CD16^{Hi} Vd2 cells expressing mesothelin-targeting CAR show potent anti-tumor effects, which is further potentiated by co-expression of IL-15, against ovarian cancer cell lines both in vitro and in vivo. In contrast to conventional alpha/beta T cells, Vd2 CAR T cells caused no xenogeneic GvHD in mice. Finally, it was demonstrated that CD16^{Hi}, but not CD16^{Lo}, Vd2 T cells show ADCC against ovarian cancer cell lines using a HER2-specific antibody. Together, this study supports the use of allogeneic CD16^{Hi} Vd2 T cells in cell therapies for cancers.

Response: Thank you for your thoughtful critique of our study. We appreciate your recognition of our methods and process employed, as well as the potent antitumor effects observed in our studies. We have carefully reviewed your comments and incorporated them into our work, and we feel they increase our work's significance and clarity.

Major comments:

4.1 One of strengths of this study is the identification of a subset of individuals who have CD16^{Hi} Vd2 T cells as donors of therapeutic Vd2 T cells. However, in contrast to detailed comparison between CD16^{Hi} and CD16^{Lo} donors, CD16^{Hi} and CD16^{Lo} Vd2 T cells within a CD16^{Hi} or CD16^{Lo} donor were not characterized. It would be important to investigate phenotype, proliferation, and anti-tumor functions of CD16^{Hi} and CD16^{Lo} Vd2 T cells within a donor.

Response 4.1: We appreciate the Reviewer's comment. In an effort to advance Vδ2 T cell therapies, we focused our investigation on bulk Vδ2 T cells expanded from each donor, as this may be the most feasible approach for off-the-shelf therapy production. We agree that it would be beneficial to investigate the differing function of CD16^{Hi} and CD16^{Lo} Vd2 T cells within a single donor. Following the Reviewer's suggestion, we have compared the expression of phenotypic markers and effector molecule (granzyme B and perforin) production of CD16^{Hi} and CD16^{Lo} Vd2 T cells from within Vδ2 T cell donors (without sorting), as shown in **Supplementary Figure 1**. We compared the expression of phenotypic markers and the production of effector molecules in CD16⁺ and CD16⁻ Vδ2 T cells within individual donors (Supplementary Figure 1). The expression of chemokine receptors CXCR3, CCR4, and CCR5 was comparable within donors and between donors, whereas CD56 was upregulated on CD16⁺ cells within donors and expressed at higher overall levels on Vδ2 T cells from CD16^{Hi} donors and CCR2 was upregulated on CD16⁻ cells within donors and expressed at overall higher levels on Vδ2 T cells from CD16^{Lo} donors. Importantly, the expression of granzyme B and perforin was similar between CD16⁺ and CD16⁻ Vδ2 T cells within a donor, and both types of cells displayed higher expression levels in Vδ2 T cells from CD16^{Hi} donors than from CD16^{Lo} donors. We have revised the manuscript to include this analysis and also added a paragraph in the results section. This supports the hypothesis that CD16 is not an active, functional receptor for increased cytotoxicity in response to tumor cells

but rather a marker for selecting donors with Vδ2 T cells possessing increased cytotoxicity. We kindly point to the Fig 2 (RNAseq results) and the Fig 1j (transgenic CD16 expression) experiments as supportive data that CD16 can be used as a biomarker to select for donors containing Vδ2 T cells with enhanced cytotoxicity following the expansion protocol used in our experiments and that the CD16 receptor itself does not functionally enhance tumor killing. Following our protocols, we do not sort Vδ2 T cells based on CD16 expression, however, we will certainly take this feedback into account for future studies and strive to provide further validation of Vδ2 T cell functionality within a donor.

4.2 In all experiments testing gamma/delta TCR-mediated anti-tumor effects, ZOL was used to sensitize cancer cells to Vδ2 T cells. Significance of TCR-mediated anti-tumor effects of Vδ2 T cells in the presence of ZOL should be explained or discussed.

Response 4.2: Following the Reviewer's suggestion, we have included a paragraph to introduce bisphosphonates and ZOL in the introduction. In addition, we have also included a new sentence in our discussion that clarifies the role of ZOL in in vitro studies and deregulated cellular metabolism presence in tumors.

4.3 Cytotoxicity of Vδ2 CAR T cells against M2-polarized macrophages is an attractive and unique attribute of Vδ2 T cells in comparison to other immune effector cells. However, only MCAR15-engineered Vδ2 T cells were tested in these experiments. It would be important to clarify roles of CAR and IL-15 in the cytotoxicity against M2 macrophages. In addition, it is important to test cytotoxicity against other myeloid cells such as M1/M0 macrophages, monocytes, and dendritic cells.

Response 4.3: We appreciate the Reviewer's comment. We previously performed a similar experiment using unmodified Vδ2 T cells (no CAR and IL-15 engineering). The data was published in *Cancers* **2022**, 14(11), 2749; <https://doi.org/10.3390/cancers14112749>. As shown in Figure 5 of the previous publication, Vδ2 T cells were capable of killing M2 macrophages without CAR and IL-15 engineering. With the addition of ZOL, the killing was further enhanced. Our previous results and our new data in Supplementary Fig 4 indicate that CAR and IL-15 are not required for M2 macrophage killing. These assays were performed using the same 24 h coculture, and the killing with and without engineering is comparable. For both engineered and nonengineered Vδ2 T cells, ZOL increases M2 macrophage killing, which implicates the Vδ2 TCR in macrophage killing. In our studies, we focus on M2 macrophages, and there is literature reporting that M0, M1, and M2 macrophages were susceptible to Vδ2 T cell cytotoxicity with the addition of ZOL (*Cancer Immunol Immunother* 66, 1205–1215 (2017). <https://doi.org/10.1007/s00262-017-2011-1>). The authors conclude that that ZOL can render M1 and M2 Mφs susceptible to Vδ2+ T cell cytotoxicity in a perforin-dependent manner, and that this has important implications regarding the use of ZA in cancer immunotherapy.

Minor comments:

4.4 In figure 3i, it appears that production of IL-15 requires co-culture with cancer cells, which is unlikely because IL-15 is co-expressed with CAR by a constitutive promoter. Is it possible

that transgenic expression of IL-15 enhances production of endogenous IL-15 production from Vd2 T cells?

Response 4.4: We have added a statement and sources to support the consistency of our findings. We have included "Marked IL-15 secretion was seen in the activated MCAR15 group (Fig. 3i) upon coculture with OVCAR3-FG cells, which is consistent with previously published data that demonstrate increased IL-15 production by CAR/IL-15 engineered $\alpha\beta$ T⁴³, V δ 1 T²⁶, and NK cells²³ following antigen stimulation. This may be due to the heightened metabolic activity and protein translation that occur during cell activation as well as the short half-life of IL-15." Given the low level of nature IL-15 by Vd2 T cells, it is unlikely that that transgenic expression of IL-15 enhances production of endogenous IL-15 production from Vd2 T cells.

4.5 For experiments of repeated stimulations (for example figure 4f-g), it is unclear whether the T cells are harvested and counted after each stimulation, so that T cell/cancer cell ratio is constant in all stimulations.

Response 4.5: We appreciate the Reviewer's comment. We have updated with more details for our "in vitro repeated tumor challenge assay". As described in our methodology, T cells are not counted after each T cell stimulation. At each round of tumor rechallenge, T cells are transferred from the old plates to the newly seeded tumor cell plates. We recognize that labs use varying in vitro repetitive tumor rechallenge assays, and we observe that protocols like our own are utilized by several CAR-T labs. We agree that maintaining a constant T cell/cancer cell ratio can be useful and also believe that the killing readout in our system is meaningful, as it shows which effector cell group can kill more rounds of tumor cells.

4.6 For supplementary fig 1, NCR2 was described in the text, but not appeared in the Figure.

Response 4.6: We have updated the text to accurately describe Supplementary Fig 1.

4.7 Figure 4i was not explained in the text.

Response 4.7: We have updated the text to properly describe Figure 4i.

4.8 The source of mesothelin specific CAR or scFv and detailed information for "preclinical anti-HER2 IgG1 analogue to trastuzumab" should be provided.

Response 4.8: We have updated the Materials section to provide more detailed information for the sources of the mesothelin specific scFv and the preclinical anti-HER2 IgG1 analogue to trastuzumab. We included "the MCAR consists of SS1 scFv, CD8 α hinge, CD28 transmembrane domain, CD28 signaling domain, and CD3 ζ signaling domain⁹³" and "InVivoSIM anti-human HER2 (Trastuzumab Biosimilar) was purchased from BioXCell (Cat. SIM0005)" in our materials and methods section.

4.9 Although the manuscript is generally well-written and straightforward, "methods" section should be carefully revised. It appears that many reagents and procedures in this section were not used in the experiments in this study. Electroporation/Nucleofection condition of "250 V for 5 minutes" is impossible. Finally, a little more detail of lentiviral vector production and

transduction should be provided: for example, packaging/envelope plasmids, timing of harvests, and a use of transduction enhancers.

Response 4.9: We have updated the Methods section to be accurate on the reagents and procedures used in this study. We have corrected our Electroporation/Nucleofection section (now included in the Cell lines section) and provided more detail on lentiviral vector production and transduction, including the packaging/envelope plasmids, timing of harvests, and a use of transduction enhancers.

REVIEWERS' COMMENTS

Reviewer #1 (Remarks to the Author):

I have read the responses to the issues raised by myself and other reviewers. I am totally satisfied in the responses to the minor issues that I had raised. In my view this paper is an important and valuable addition to the field

Reviewer #2 (Remarks to the Author):

The authors addressed the issues raised appropriately. I am satisfied with the improvements and updates the authors have made in response to my previous comments and suggestions.

In particular, I appreciate the addition of the comparison of cytotoxicity based on CD16 expression in Figure R1. This supports the linear relationship between CD16 expression and the tumor killing activity of V δ 2 T cells. As shown in Figure R2, the ADCC studies using anti-CD20 mAb provide valuable insight into the potential applicability of this approach. I agree with the authors' explanation of the differences between IL-5 combined with CAR T cells and IL-5 transduced CAR T cells, as well as the discussion of the limitations and challenges in the research. These are well presented and increase the overall quality of the article.

Additionally, the clarification regarding the MSLN knockout and parental OVCAR3-FG cell lines in Figure 4 and the revised figure legends in Figure 5c provide better context and understanding of the results. I am also pleased to see the inclusion of the possible limitations of tumor penetration in the discussion, as well as the acknowledgment of the potential challenges and adverse effects of engineered CD16Hi V δ 2 T cell-based therapies.

Considering the comprehensive and satisfactory revisions the authors have made, I think the manuscript deserves publication.

Reviewer #4 (Remarks to the Author):

This reviewer appreciates comprehensive responses that sufficiently address the comments raised by this reviewer. There is no additional comment from this reviewer.